# The Progress of Cobalt-Based Anode Materials for Lithium Ion Batteries and Sodium Ion Batteries

**Yaohui Zhang [1],\*, Nana Wang [2] and Zhongchao Bai [2],\***

[1]   School of Energy and Power Engineering, North University of China, Taiyuan 030051, China
[2]   College of Sciences, Henan Agricultural University, Zhengzhou 450002, China
\*   Correspondence: zhangyaohui@nuc.edu.cn (Y.Z.); baizhongchao@tyut.edu.cn (Z.B.)

**Abstract:** Limited by the development of energy storage technology, the utilization ratio of renewable energy is still at a low level. Lithium/sodium ion batteries (LIBs/SIBs) with high-performance electrochemical performances, such as large-scale energy storage, low costs and high security, are expected to improve the above situation. Currently, developing anode materials with better electrochemical performances is the main obstacle to the development of LIBs/SIBs. Recently, a variety of studies have focused on cobalt-based anode materials applied for LIBs/SIBs, owing to their high theoretical specific capacity. This review systematically summarizes the recent status of cobalt-based anode materials in LIBs/SIBs, including $Li^+/Na^+$ storage mechanisms, preparation methods, applications and strategies to improve the electrochemical performance of cobalt-based anode materials. Furthermore, the current challenges and prospects are also discussed in this review. Benefitting from these results, cobalt-based materials can be the next-generation anode for LIBs/SIBs.

**Keywords:** cobalt-based materials; anode; lithium ion battery; sodium ion battery

## 1. Introduction

With the increasing consumption of fossil energy, more and more attention has been paid to the development and utilization of renewable energy. However, new energy resources like wind, solar and tidal power are not sustainable by virtue of their uncertainty over time. In this scenario, the development of high-performance energy storage devices is highly necessary. To this end, rechargeable batteries (RBs) are of particular interest due to the multiple advantages of rational working voltage, high capacities and long cycling life [1–4]. Lithium ion batteries (LIBs), as one of the dominant RBs, have gradually penetrated into many aspects of our lives, such as portable electronic devices [5], consumable electronics [6] and electric vehicles [7]. However, the large-scale application of LIBs is hobbled by the rareness of lithium resources in the earth's crust [8,9]. Sodium ion batteries (SIBs) have captured great concerns as a complementary technology for LIBs by virtue of abundant raw materials. Furthermore, many electrode materials of LIBs can be applied as a drop-in replacement for SIBs because of a similar conversion reaction mechanism with LIBs.

There is a growing demand for the energy density of RBs. For LIBs, the mainstream anode materials-graphite materials are unable to meet the ever-increasing requirement of energy density because of the relatively lower theoretical specific capacity (~372 mA h $g^{-1}$). As for SIBs, the larger radius of $Na^+$ (1.02 Å vs. 0.76 Å of $Li^+$) gives rise to severe volume fluctuations of electrode materials and sluggish diffusion kinetics, finally yielding poor electrochemical performances of SIBs. Up to now, there have been no right anode materials of SIBs for practical applications [10]. Therefore, a key requirement for both LIBs and SIBs is the development of anode materials with applicable working voltage, marvelous cycle performance and high energy density.

Up to now, unceasing efforts have been devoted to exploring appropriate anode materials, like metal chalcogenides, alloying metals, metal phosphide and carbon materials. Among these materials, cobalt-based composites (oxides, sulfides, phosphides and alloys) have recently drawn enormous attention owing to their outstanding electrochemical performances. For example, Zhang et al. synthesized a porous fiber structured $Co_3O_4$@carbon for LIBs that delivered a capacity of 558 mA h $g^{-1}$ after 500 cycles at a current density of 5 A $g^{-1}$ [11]. Bai's group fabricated a spongy $CoS_2$/carbon composite that showed marvelous electrochemical performances as anode materials for both LIBs and SIBs. It retained a capacity of 610 mA h $g^{-1}$ after 120 cycles at 0.5 A $g^{-1}$ in LIBs and 330 mA h $g^{-1}$ after 60 cycles at 0.5 A $g^{-1}$ in SIBs, respectively [12]. Zhang and his colleagues synthesized CoP nanoparticles embedded in N-doped carbon nanosheets that displayed a Na-storage capacity of 598 mA h $g^{-1}$ at 0.1 A $g^{-1}$ [13].

Nonetheless, a large proportion of cobalt-based electrode materials are subjected to severe volume fluctuations caused by the uptaking and releasing of $Li^+$ or $Na^+$, which eventually yield a pulverization of electrode materials and detachment from the current collector, finally giving rise to a decaying of capacity [14–16]. To circumvent these issues, plenty of cobalt-based materials with special structures have been reported [17–20]. In general, active materials with a nanometer scale can be conducive to the transfer of $Li^+$ or $Na^+$ owing to shorter diffusion lengths and tolerate the strain caused by volume expansion during periodical cycle processes, consequently enhancing electrochemical performances [21,22]. Zhang et al. fabricated $Co_9S_8$@carbon nanospheres with an average size of 60 nm, which delivers an astonishing rate capability of 305 mA h $g^{-1}$ after 1000 cycles at 5 A $g^{-1}$ for anode materials of SIBs [23]. $CoSe_2$/N-doped carbon nanofibers synthesized by Li's group shows excellent structure retention. Even cycled for 1000 cycles, the sample well maintained its original structure without pulverization or amorphization [24].

Furthermore, the electrochemical properties of cobalt-based materials can also be improved by combining with carbonaceous buffers [19,25,26]. Beneficially, the carbonaceous buffer can enhance the electrical conductivity of the composite, as well as mitigate pulverization and aggregation during periodical cycles. Ge et al. synthesized unique CoP@C-reduced graphene oxide-nickel foam composite as the anode for SIBs that delivered a splendid capacity of 473.1 mA h $g^{-1}$ at 0.1 A $g^{-1}$ after 100 cycles [27]. $Co_9S_8$ embedded in N-rich carbon hollow spheres fabricated by Fang's group exhibited a discharge capacity of 518 mA h $g^{-1}$ at a current density of 2.176 A $g^{-1}$ for LIBs [28]. To carefully study the origin behind the marvelous storage properties, many advanced testing technologies like in-situ X-ray diffraction, in-situ Nyquist plots, in-situ transmission electron microscopy and first-principles calculations are applied to discover the Li/Na-ion storage mechanism in cobalt-based electrode materials containing morphological evolution, phase transformation, kinetics, etc. These findings will construct a theoretical basis for a more applicable design of cobalt-based materials as an anode for LIBs and SIBs [29–33].

Until now, a lot of reports on cobalt-based electrode materials have been published. However, there is no review focused on a cobalt-based anode for LIBs/SIBs. In this review, we summarized the recent development of cobalt-based anode materials for LIBs and SIBs, including electrochemical mechanisms, synthetic strategies and common applications. Moreover, we also remarked on the future prospects of cobalt-based anode materials within the rapidly developing energy storage field.

## 2. $Li^+$/$Na^+$ Storage Mechanism

$Li^+$/$Na^+$ storage mechanisms for different Co-based materials are shown in Figure 1. Co-based electrode materials can be roughly divided into three types: Co-based alloys [19,34]; compounds of cobalt and nonmetals ($Co_xA_y$, A=O, P and S . . . ) [18,35–38] and other composites fabricated by cobalt and metal nonmetals [39]. Unlike graphite-based materials, $Li^+$/$Na^+$ storage mechanisms of cobalt-based electrode materials consist mainly of alloying and conversion reactions. During the processes of alloying and conversion reactions, atomic rearrangements occur along with structural evolution and phase transformation, thereby yielding severe volume fluctuations during battery cycles.

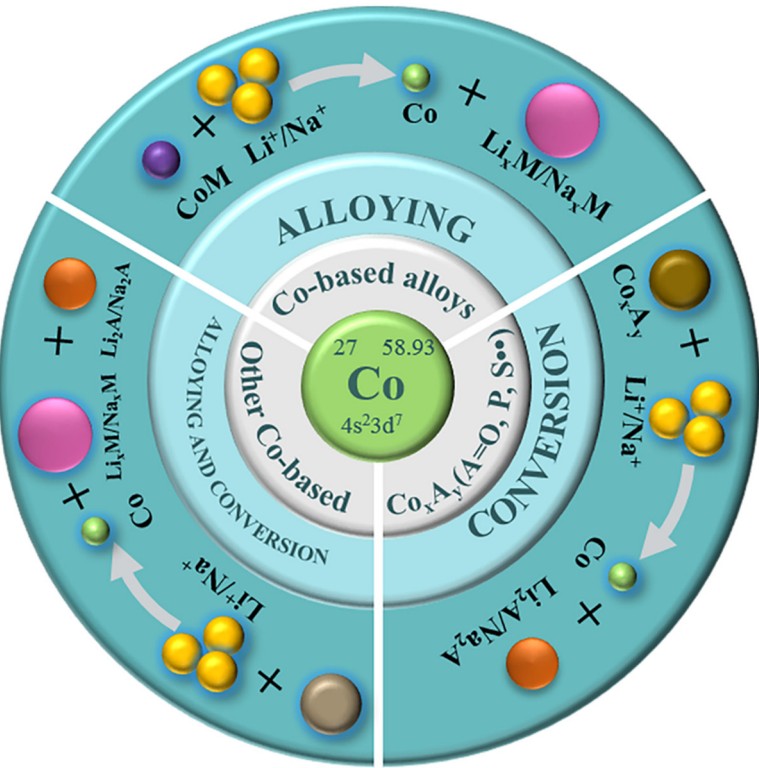

**Figure 1.** $Li^+/Na^+$ storage mechanisms for different Co-based materials.

### 2.1. $Li^+/Na^+$ Storage Mechanism in Co-Based Alloys

For cobalt-based alloys, lithium/sodium ion storage properties are decided by metal components that make up the alloy with cobalt [40–43]. Even so, the presence of cobalt components in the alloys is essential, which disperses into the alloys to improve the conductivity of alloys, as well as the retard volume expansion during cycles [44,45]. Many types of research on alloying storage mechanisms of cobalt-based materials have been published. For example, Yu et al. briefly pointed out the lithium ion storage mechanisms of Sn-Co@polymethylmethacrylate (PMMA) nanospheres [40]. In Figure 2a, an obvious slope between 1.2 and 0.4 V during the first discharge can be ascribed to the formation of SEI film. Then, a plat plateau at 0.3 V is attributed to the initial lithiation of the Sn component of the alloy, including the isolation of Co and the formation of $Li_xSn$. Subsequently, the platform centered at about 0.5 V during the charge can be ascribed to the delithiation of $Li_xSn$ and the formation of Sn. The entire lithium storage process of the alloy can be described by following equations:

$$Co_ySn + xLi^+ + xe^- \rightarrow Li_xSn + yCo(0 < x < 4.4) \text{ Discharge} \tag{1}$$

$$Li_xSn + yCo \rightarrow Sn + yCo + xLi^+ + xe^- \text{ Charge} \tag{2}$$

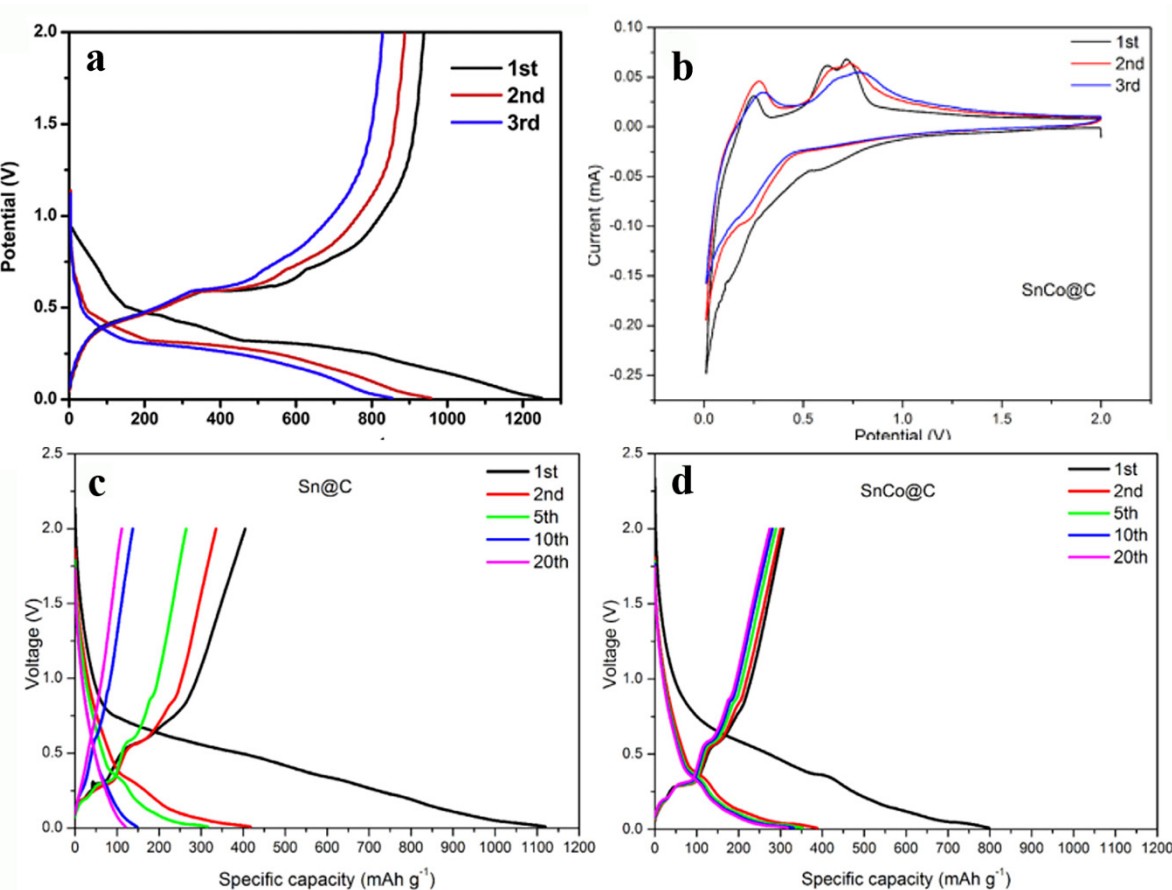

**Figure 2.** Li$^+$ and Na$^+$ storage in the Sn-Co alloy anode. (**a**) Charge-discharge profiles of Sn-Co@polymethyl methacrylate (PMMA) for the initial three cycles in lithium ion batteries (LIBs) [40]. (**b**) Cyclic voltammetry (CV) curves of SnCo@C in sodium ion batteries (SIBs) for the initial three cycles [41]. (**c**) and (**d**) Charge-discharge curves of Sn@C and SnCo@C at different cycles in SIBs, respectively [41].

As displayed in Figure 2b, the sodium storage mechanism of cobalt alloy can be simply demonstrated by cyclic voltammetry (CV) curves of SnCo@C in SIBs for the initial three cycles [41]. During the first discharge process, peaks centered at around 0.75 and 0.15 V can be ascribed to the first sodiation of the alloy, which indicated the formation of Na$_x$Sn and Co. The peaks at 0.3, 0.6 and 0.7 V upon the charge process represented a desodiation process from Na$_x$Sn to Sn. Similar to the lithium storage mechanism, the sodium storage mechanism of SnCo@C can be concluded by following equations:

$$SnCo + xNa^+ + xe^- \rightarrow Na_xSn + Co(0 < x < 4.4) \text{ Discharge} \tag{3}$$

$$Na_xSn + Co \rightarrow Sn + Co + xNa^+ + xe^- \text{ Charge} \tag{4}$$

In Figure 2c,d, the charge-discharge profiles of Sn@C and SnCo@C displayed similar voltage platforms at different cycles, indicating the decisive role of the metal component alloyed with cobalt. Furthermore, the presence of cobalt improves the capacity retention ratio during cycles, suggesting the importance of cobalt.

### 2.2. Li$^+$/Na$^+$ Storage Mechanism in Co$_x$A$_y$ (A=O, S, P and Se)

Co$_x$A$_y$ electrode materials consist of cobaltous oxides, cobaltous sulfides, cobaltous phosphide and cobaltous selenide. Li$^+$/Na$^+$ storage mechanism in Co$_x$A$_y$ is the conversion reaction that eventually



forms Co and $Li_xA$ or $Na_xA$ [14,15,20,27,35,46–48]. The typical reactions of $Co_xA_y$ with $Li^+$ is displayed in Table 1. It should be noted that some temporary intermediate reactions may occur during the discharge process of $Co_xA_y$, as depicted in the following equations:

$$Co_xA_y + Li^+ + e^- \rightarrow LiCo_xA_y \tag{5}$$

or

$$Co_xA_y + Li^+ + e^- \rightarrow Co_xA_{y-1} + LiA \tag{6}$$

**Table 1.** $Li^+$ storage mechanism of some $Co_xA_y$-active materials.

| Active Materials | $Li^+$ Storage Mechanism | Reference |
|:---:|:---:|:---:|
| $Co_3O_4$ | $Co_3O_4 + 8Li^+ + 8e^- \rightarrow 3Co + 4Li_2O$ | [15] |
| $Co_2P$ | $Co_2P + 3Li^+ + 3e^- \rightarrow 2Co + Li_3P$ | [46] |
| $CoS_2$ | $CoS_2 + 4Li^+ + 4e^- \rightarrow Co + 2Li_2S$ | [14] |
| $CoSe_2$ | $CoSe_2 + 4Li^+ + 4e^- \rightarrow Co + 2Li_2Se$ | [20] |

Furthermore, the sodium storage reactions are similar to lithium in terms of the conversion reaction accompanied by temporary intermediate reactions. Zhang's group (Figure 3a) investigated the sodium storage behavior of shale-like $Co_3O_4$ by cyclic voltammetry (CV) from 0.005 to 2.9 V at 0.1 mV/s [35]. During the first discharge process, they found an intermediate reaction initially happened at 0.76 V, with $Co_3O_4$ reduced to CoO and $Na_2O$, provisionally. Subsequently, the resulting CoO intermediate phase was further converted (0.39 V) to metallic Co and $Na_2O$. Upon oxidation, metallic Co was step-by-step oxidized from CoO to $Co_3O_4$ with the extraction of $Na^+$ from $Na_2O$. Two main anodic peaks centered at 0.88 and 1.65 V corresponding to the reversible formation of CoO and $Co_3O_4$, respectively. The sodium storage reactions of the shale-like $Co_3O_4$ are as follows:

$$Co_3O_4 + 2Na^+ + 2e^- \rightarrow 3CoO + Na_2O \tag{7}$$

$$3CoO + 6Na^+ + 6e^- \rightarrow 3Co + 3Na_2O \tag{8}$$

Ge et al. investigated the $Na^+$ storage mechanism of CoP by using ex-situ TEM and CV tests [27]. In Figure 3b, two peaks centered at 0.2 and 0.05 V can be attributed to the stepwise formation of $Na_xP$ and $Na_3P$ during the first cathodic scan. During the subsequent charge process, a stepwise extraction of $Na^+$ occurred along with the decomposition of $Na_3P$, characterized by two anodic peaks located at 0.5 and 1.5 V. It should be noted that a partial $Na_3P$ transformed to P during the decomposition of $Na_3P$. The corresponding sodiation/desodiation reactions are as follows:

$$CoP + 3Na^+ + 3e^- \rightarrow Co + Na_3P \tag{9}$$

$$Na_3P \rightarrow P + 3Na^+ + 3e^- \tag{10}$$

$$P + 3Na^+ + 3e^- \leftrightarrow Na_3P \tag{11}$$

Fei and coworkers investigated the electrochemical reactions of $CoS_2$/rGO by CV curves of the initial two cycles (Figure 3c) [49]. The peak located at 1.4 V can be ascribed to the intercalation of $Na^+$ into the $CoS_2$ crystals in the first discharge process. The resulting intermediate $Na_xCoS_2$ suffered a further conversion reaction to form $Na_2S$ and metallic Co at 0.8 V. Upon desodiation, two main oxidation peaks at 1.6 and 2.2 V can be due to the reversible formation of $Na_xCoS_2$ and $CoS_2$, respectively. The corresponding electrochemical reactions are as follows:

$$CoS_2 + xNa^+ + xe^- \leftrightarrow Na_xCoS_2 \tag{12}$$

$$Na_xCoS_2 + (4-x)Na^+ + (4-x)e^- \leftrightarrow Co + 2Na_2S \tag{13}$$

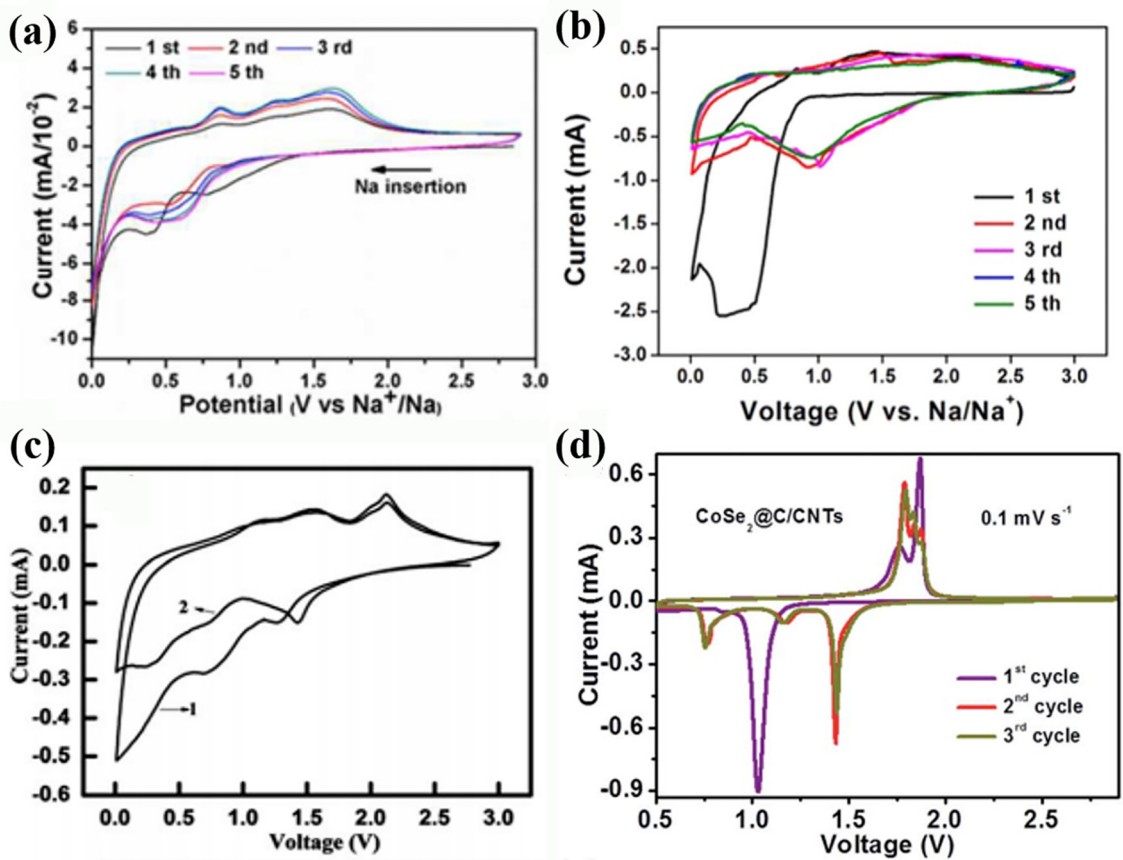

**Figure 3.** CV curves of (**a**) shale-like $Co_3O_4$ tested at 0.1 mV/s from 0.005 to 2.9 V vs. Na/$Na^+$ [35]. (**b**) CoP@C-RGO-NF tested at 0.1 mV/s in the voltage range of 0.01–3 V vs. Na/$Na^+$ [27]. (**c**) $CoS_2$/rGO at a scan rate of 0.5 mV/s from 0.01–3 V vs. Na/$Na^+$ [49]. (**d**) $CoSe_2$@C/carbon nanotubes (CNTs) tested at 0.1 mV/s from 0.5–2.9 V [48].

The sodium storage reaction mechanism of $CoSe_2$ was investigated by Qiu's group with a CV test, as shown in Figure 3d [48]. During the first anodic scan, two adjacent peaks at 1.7 and 1.87 V were attributed to stepwise conversion reactions from Co and $Na_2Se$ to an intermediate $Na_xCoSe$ and a final $CoSe_2$, respectively. In the second anodic scan, three reduction peaks appeared at 1.4, 1.1 and 0.68 V, corresponding to the uptake of $Na^+$ and a stepwise sodiation of $CoSe_2$. The corresponding reactions can be seen as follows:

Discharge process:

$$CoSe_2 + xNa^+ + xe^- \rightarrow Na_xCoSe_2 \tag{14}$$

$$Na_xCoSe_2 + (2-x)Na^+ + (2-x)e^- \rightarrow CoSe + Na_2Se \tag{15}$$

$$CoSe + 2Na^+ + 2e^- \rightarrow Co + Na_2Se \tag{16}$$

Charge process:

$$Co + 2Na_2Se \rightarrow Na_xCoSe_2 + (4-x)Na^+ + (4-x)e^- \tag{17}$$

$$Na_xCoSe_2 \rightarrow CoSe_2 + xNa^+ + xe^- \tag{18}$$

The Li$^+$/Na$^+$ storage mechanism is decided by components that composited with cobalt. Similarly, other cobalt-based composites like Co$_2$SnO$_4$, CoMn$_2$O$_4$ and Sb-Co-P have more complex Li$^+$/Na$^+$ storage mechanisms, due to more components in composites [39,50–52]. Nevertheless, the Li$^+$/Na$^+$ storage mechanisms are still composed of basic alloying and conversion reactions.

## 3. Preparation Methods of Synthesizing Cobalt-based Active Materials

Up until now, many reports have been published for different synthetic strategies of various cobalt-based anode materials. In this section, we review some common methods.

### 3.1. Hydrothermal/Solvothermal Methods

As a typical synthetic method, hydrothermal/solvothermal reactions are mainly used to prepare materials by providing appropriate temperatures and times [53]. Under the circumstances of moderate temperatures and high pressures, various structured cobalt-based materials have been synthesized with high-phase purity and crystallinity. Moreover, hydrothermal/solvothermal methods are environmentally benign and cost-efficient due to their own characteristics [21,54]. Many cobalt-based anode materials with various morphologies have been synthesized by these methods, like nano-sized Co-Sn alloy, single-crystal intermetallic Co-Sn nanospheres and single-crystalline Co$_3$O$_4$ nanobelts [55–57].

Zhang's group synthesized a facile structure with CoS$_2$ confined in the graphitic carbon walls of porous N-doped carbon spheres (CoS$_2$-in-wall-NCSs) by the hydrothermal method associated with carbonization and sulfurization, as shown in Figure 4a–c [58]. During the process of hydrothermal, Co-containing melamine-phenolic resin spheres formed under the condition of high temperatures and high pressures. Followed with carbonization and sulfurization, CoS$_2$ nanodots were evenly distributed throughout the carbon sphere. Figure 4d,e reveals the worm-like sample with CuCo$_2$S$_4$ nanocrystalline (5–20 nm) evenly anchored on the carbon nanotubes (CNTs) synthesized by Jin et al. through solvothermal treatment [59]. CoSe@ carbon spheres (denoted as CoSe@CSs) were prepared through a facile solvothermal treatment by adding Co(NO$_3$)$_2$·6H$_2$O and H$_3$O$_2$Se to a moderate isopropanol solution [60]. The TEM image (Figure 4f) reveals that CoSe nanoparticles uniformly embedded in amorphous carbon spheres (100 nm). Zhu's group synthesized a pure orthorhombic CoP film of nanorod arrays through a hydrothermal method [61]. As displayed in Figure 4g, these CoP nanorods were evenly distributed and interconnected with others to form a 3D interwoven. Li and coworkers reported a facile solvothermal reaction to synthesize Co$_3$O$_4$/CNT nanocomposites [62]. In Figure 4h, the cyboidal Co$_3$O$_4$ nanoparticles were evenly anchored on the surface of the CNT. The elemental mapping of the sample reveals the distribution of C, Co and O.

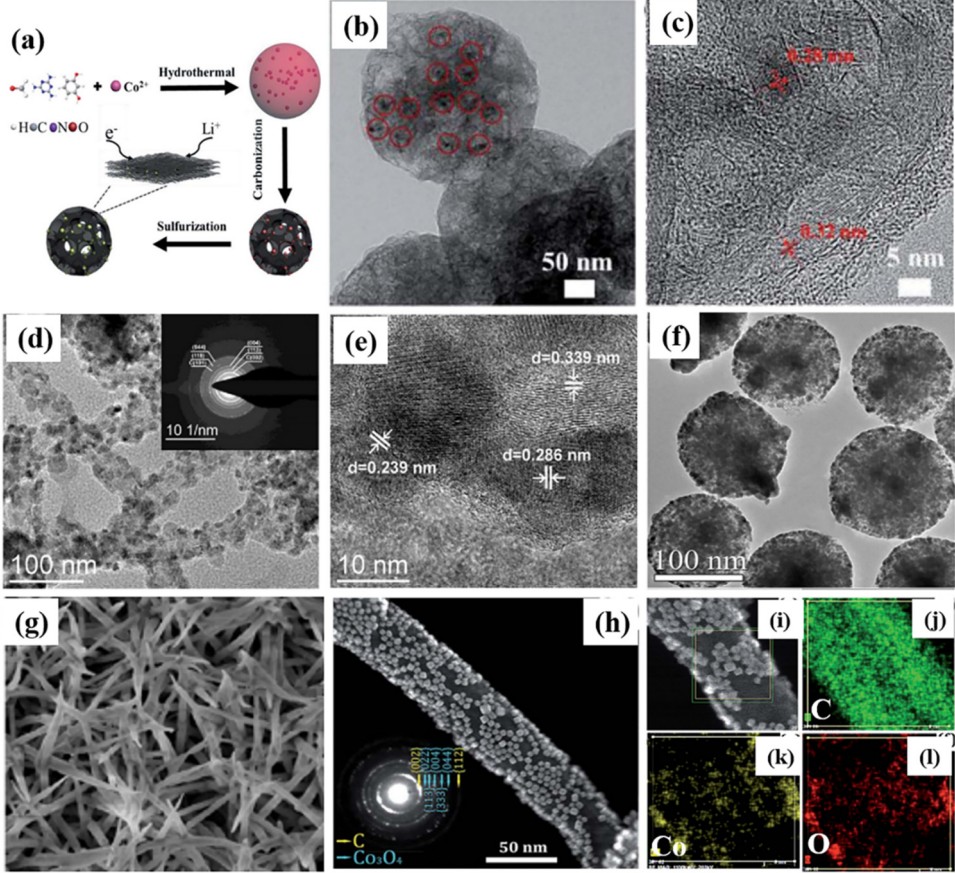

**Figure 4.** Preparation methods and morphology characterization of cobalt-based composites. (**a**) Schematic illustration of the preparation of (**b**) TEM and (**c**) HRTEM images of the CoS₂-in-wall-N-doped carbon spheres (NCSs) [58]. (**d**) TEM image, SAED pattern (inset) and (**e**) HRTEM image of CNTs@CuCo₂S₄ [59]. (**f**) Low-magnification TEM image of CoSe@100CSs [60]. (**g**) SEM image of CoP nanorod arrays [61]. (**h**) Low and (**i**) high-magnification SEM images and (**i**–**k**) elemental mapping images of Co₃O₄/CNT nanocomposites [62].

## 3.2. Galvanic Replacement

Galvanic replacement has been widely applied to synthesize nanostructured electrode materials due to its facile and low-temperature synthesis method. Zhou and coworkers synthesized a broccoli-like Co₃O₄@CNTs anode material for SIBs through a galvanic replacement route [63]. Firstly, the acidizing operation of carbon nanotubes (CNTs) can create numerous functional groups with a negative charge on the surface of CNTs. Owing to electrostatic attraction, lots of Co²⁺ ions with a positive charge will attach to the surface of CNTs with carboxylic groups and serve as nucleation precursors. During the subsequent galvanic replacement reaction, a Co²⁺ ion was oxidized to Co₃O₄ at 140 °C. After that, Co₃O₄ crystalline particles aggregated on the surface of CNTs owing to the electrostatic attraction and Van der Waals forces; thus, broccoli-like Co₃O₄@CNTs were finally obtained (Figure 5a). The TEM and HRTEM images displayed in Figure 5b–d clearly confirmed that the well-crystallized Co₃O₄ nanocrystals were assembled on the surface of the CNTs. Ma and collogues successfully synthesized a novel Sn-Co@C alloy through galvanic replacement with metal-organic framework ZIF-67 as both the template and carbon source [64]. Figure 5e–g reveal that the Sn-Co nano-alloy particles (~10 nm) were evenly embedded in porous N-doped carbon, forming a novel hierarchical structure. In Figure 5h–l, the distribution of C, O, N, Co and Sn elements clearly confirms the uniform structure of the Sn-Co@C alloy.

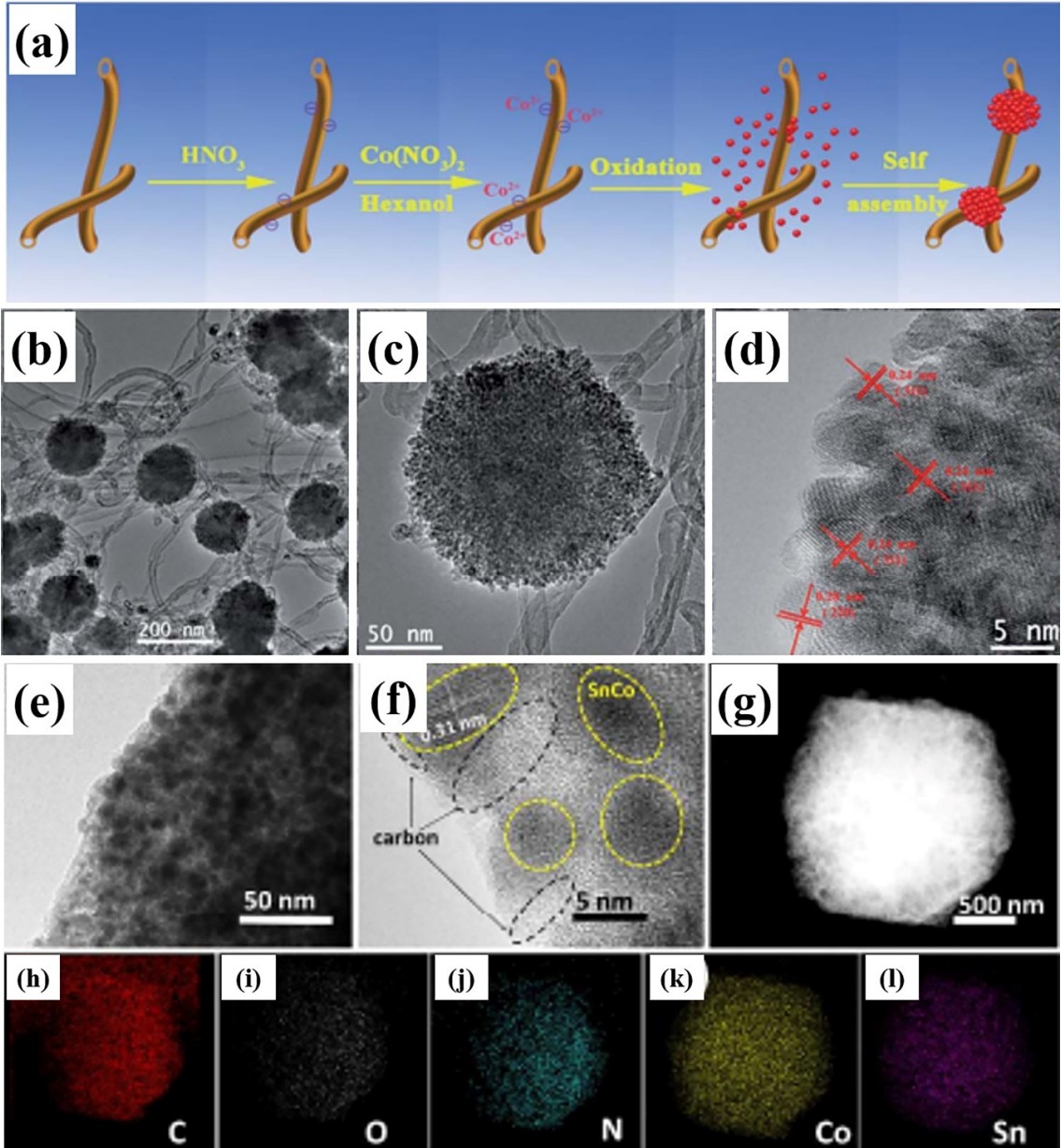

**Figure 5.** Morphology characterization of cobalt-based composites. (**a**) Schematic evolution of the $Co_3O_4$@CNT composites. TEM images (**b**,**c**) and HRTEM image (**d**) of the $Co_3O_4$@CNT composites [63]. TEM (**e**) and HRTEM (**f**) images of Sn-Co@C-2; STEM image (**h**) and corresponding elemental mapping images (**h**–**l**) of Sn-Co@C-2 [64].

Mesoporous $Co_3O_4$ nanowire arrays, needlelike $Co_3O_4$ nanotubes [65], porous carbon/$CoS_2$ [66] and some other cobalt-based anode materials with various morphologies and structures have also been synthesized by galvanic replacement [67].

### 3.3. Heat Treatment

As a high-temperature treatment, heat treatments can be used independently and combined with other synthesis methods, such as ball milling, electrostatic spinning and hydrothermal/solvothermal, to provide a special reaction condition for intermediate processes like sulfuration and carbonization. Lots of cobalt-based electrode materials with high performances have been successfully synthesized through heat treatments [15,22,68–79].

Pan's group successfully distributed CoSe nanoparticles into the porous carbon polyhedral (CoSe@PCP) through a two-step heat treatment [68]. During the first heat treatment, the precursor ZIF-67 was firstly converted to Co@PCP, including the carbonization of the organic linkers and reduction of the Co ion. Subsequently, CoSe@PCP was finally synthesized by selenization of Co@PCP at a high temperature (Figure 6a). As shown in Figure 6b, the resulting CoSe powders were uniformly embedded into porous carbon without aggregation. Zhang and coworkers prepared a 3D spongy $CoS_2$/C anode material through a heating precursor accompanied by a hydrothermal process [12]. Figure 6c delivers a TEM image of the synthesized spongy $CoS_2$/C with $CoS_2$ nanoparticles (~20 nm) homogeneously distributed in the porous carbon. Wang and colleagues realized accurate control over the synthesis of multi-shelled $Co_3O_4$ hollow microspheres by adjusting the solution and subsequent heating in the air [69]. The TEM and HRTEM images displayed in Figure 6d,e clearly reveal the morphology of triple-shelled $Co_3O_4$ hollow microspheres. Yin's group firstly synthesized a unique core-shell porous FeP@CoP phosphide microcube interconnected by reduced graphene oxide nanosheets (RGO@CoP@FeP) through a facile phosphorization process [70]. In Figure 6f, the SEM image of RGO@CoP@FeP reveals a typical hierarchical structure of wrinkled RGO interconnected microcubes. Qiu's group synthesized micro-scaled spherical $CoSn_2$/Sn alloy through a sintering process, employing a stoichiometric amount of $SnO_2$, $Co_3O_4$ and C as raw materials [22]. SEM observation of $CoSn_2$/Sn alloy reveals that the particles were composed of lots of small grains, as shown in the inset of Figure 6g. Zeng et al. developed a facile heat treatment, composed of pyrolysis and sulfurization, to synthesize hollow $Co_9S_8$/N-C composites [28]. The SEM image of hollow $Co_9S_8$/N-C composites displayed in Figure 6h reveals the thickness of the wall was approximately 70 nm. Co/(NiCo)$Se_2$ box-in-box structures were firstly fabricated by Kang's group via a selenization process at $Ar/H_2$, applying well-known ZIF-67 as a template [71]. As shown in Figure 6i, the thickness of the outer Ni-Co selenide shell is about 60 nm, while the inner $CoSe_2$ shell shows a very uneven thickness.

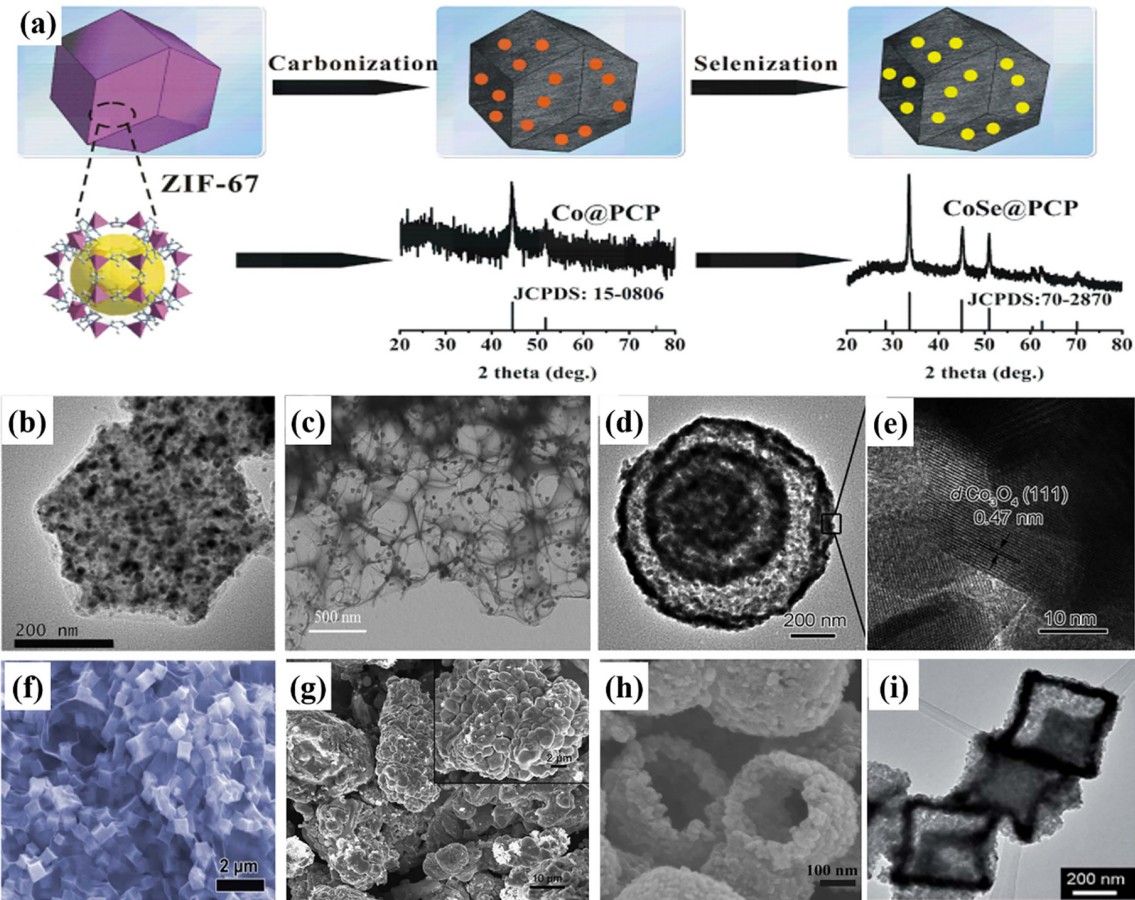

**Figure 6.** Preparation methods and morphology characterizations of cobalt-based composites. (**a**) Schematic illustration of the preparation of CoSe@porous carbon polyhedral (PCP). (**b**) TEM image of CoSe@PCP [68]. (**c**) TEM image of 3D spongy $CoS_2$/C [12]. (**d**) TEM and (**e**) HRTEM images of triple-shelled $Co_3O_4$ hollow microspheres [69]. (**f**) SEM image of the reduced graphene oxide nanosheet (RGO@CoP@FeP) composite [70]. (**g**) SEM images of the $CoSn_2$/Sn alloy [22]. (**h**) SEM image of the $Co_9S_8$/N-C hollow spheres [28]. (**i**) TEM image of the Co/(NiCo)$Se_2$ nanocubes with a box-in-box structure [71].

### 3.4. High-Energy Mechanical Milling

As a facile preparation technology, high-energy mechanical milling (denoted as HEMM) has also been commonly applied to the fabrication of cobalt-based materials owing to its large-scale production [34,42,45,80–83]. For HEMM, the products are synthesized under high temperatures and pressures generated by highly energetic collisions of tiny balls.

Carbon-coated $CoP_3$ ($CoP_3$@C) nanocomposites were synthesized by HEMM methods with a certain proportion of phosphorus, cobalt powders and carbon black [80]. The synthesis of the schematic illustration is displayed in Figure 7a. HRTEM images shown in Figure 7b reveal the presence of $CoP_3$ nanoparticles and a carbon matrix. B. Scrosati's group successfully transformed a certain proportion of Sn, Co and graphite into a nanosized SnCoC-2 alloy through HEMM [34]. Figure 7c shows the SEM image of the obtained SnCoC-2 alloy. Li et al. prepared $Sn_{30}Co_{30}C_{40}$ electrode materials by mechanical attrition with $CoSn_2$, Co and graphite as the precursors [81]. SEM image of the obtained $Sn_{30}Co_{30}C_{40}$ (Figure 7d) reveals that the average size of $Sn_{30}Co_{30}C_{40}$ particles is about 1 μm.

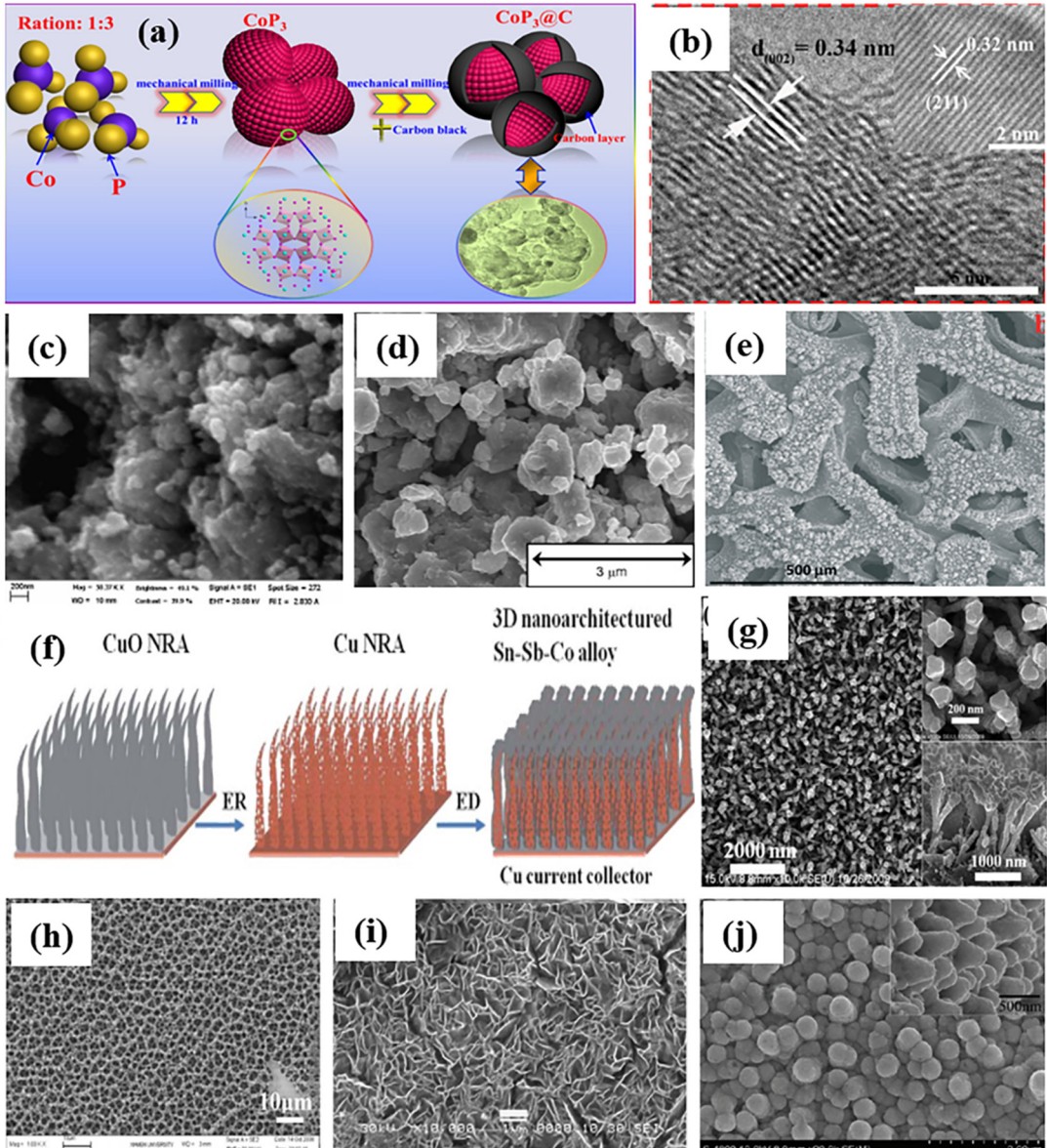

**Figure 7.** Preparation methods and morphology characterizations of cobalt-based composites. (**a**) Schematic illustration of the preparation of $CoP_3$@C microspheres. (**b**) HRTEM image of the $CoP_3$@C composite [80]. (**c**) SEM image of the SnCoC-2 sample [34]. (**d**) SEM image of the $Sn_{30}Co_{30}C_{40}$ material [81]. (**e**) SEM image of the Co-Sn alloy after electrodeposition [84]. (**f**) Schematic illustration of the preparation of the Sn-Sb-Co alloy. (**g**) SEM images of the SnSbCo-3 electrode; the insets show a corresponding high-magnification image (top-right) and cross-sectional photos of the electrode (bottom-right) [85]. (**h**) SEM image of the porous SnCo alloy [86]. (**i**) SEM image of $Co_3O_4$ thin films synthesized by electrochemical deposition [87]. (**j**) SEM image of the Sn-Co alloy deposited on a Ni nanocon-array [38].

## 3.5. Electrodeposition

Electrodeposition, as a method for separating metals or alloys from their compounds, has been used in the preparation of cobalt-based electrode materials [18,38,43,84–89]. Ricardo's group prepared the Co-Sn alloy by depositing Sn and Co atoms through the co-electroplating method, according to moderate time and current density [84]. Figure 7e indicates that the deposited Co-Sn alloy covered the nickel substrate with certain roughness and protuberances. Sun's group reported a novel nanoarchitectured Sn-Sb-Co alloy through direct electrodeposition on a Cu nanoribbon array [85].

A schematic diagram of the preparation of 3D nanoarchitectured Sn-Sb-Co alloy is shown in Figure 7f. SEM images (Figure 7g) clearly reveal the 3D array nanostructure and thickness (1500 nm) of the Sn-Sb-Co alloy.

A porous Sn-Co alloy was fabricated by Sun's group via electrodepositing the Sn-Co ally on the porous Cu film prepared by electroless plating [86]. SEM image (Figure 7h) indicates that the Sn-Co alloy was uniformly covered on the porous Cu film. Porous $Co_3O_4$ nanostructured thin films were synthesized via a facile electrodepositing method (Figure 7i) [87]. SEM image reveals that the sample was composed of small nanoflakes of about 30–40 nm in thickness. Du et al. successfully a synthesized nanoarchitectured Sn-Co alloy electrode through a two-step electrodeposition [38]. The SEM image in Figure 7j evidently indicates that there is sufficient room between the Sn-Co cylinders

### 3.6. Other Methods

With the exception of these methods, many other practical approaches, such as electrospinning, magnetron sputtering, chemical vapor deposition (CVD) and electric arc melting, have been applied to synthesize cobalt-based electrode materials with novel nanostructures [19,44,50,90–96].

Electrospinning is a method to synthesize 1D nanomaterials by the breakdown of a high-polymer solution by a high-voltage electrostatic field [24,93,97]. Li's group distributed $CoSe_2$ nanoparticles uniformly into nitrogen-doped carbon nanofibers through two steps, including the electrospinning approach and subsequent selenization process (Figure 8a) [24]. TEM image, HRTEM image and corresponding elements mapping clearly reveal that $CoSe_2$ particles were well-wrapped in the N-doped nanofibers without aggregation, as shown in Figure 8b-d, respectively. Co-Sn/CNF composites were fabricated by Lee's group through electrospinning, followed with heat treatments [90]. SEM image of the Co-Sn/CNF surface shows that the average diameter of the fibers was 180 nm (Figure 8e). Wang et al. prepared carbon-encapsulated wire-in-tube $Co_3O_4/MnO_2$ heterostructure nanofibers through electrospinning, followed by calcination [91]. The TEM image in Figure 8f shows a tubular structure with three layers, including an $MnO_2$ wire, $Co_3O_4$ tube and carbon layer, subsequently. X-ray EDS in Figure 8g clearly indicates the presence of Mn, Co, O, N and C elements. Kim's group prepared 1D carbon nanofibers embedded with uniformly SnCo nanoparticles via electrospinning, followed by calcination [92]. SEM image in Figure 8h shows that SnCo alloys with size distributions of 5–10 nm were homogeneously encapsulated in carbon nanofibers. Niu's group successfully synthesized a novel urchin-like sample composed of $CoSe_2$ nanofibers rooted into the carbon nanofibers ($CoSe_2$@CNFs) [93]. In the preparation process, Niu's group firstly synthesized the precursor (Co@CNF) via electrospinning, followed by calcination, and subsequently prepared $CoSe_2$@CNFs through a hydrothermal selenation. TEM image and corresponding HRTEM image (inset) in Figure 8i obviously show that urchin-like $CoSe_2$ nanorods with lengths of 50–100 nm and diameters of 20–30 nm are rooted into the electrospun CNFs. Qiu et al. employed the electrospinning method to synthesize $Co/Co_3O_4$-carbon nanofibers with porous structures [94]. The $Co/Co_3O_4$ nanoparticles were distributed throughout the carbon nanofiber, as shown in Figure 8j.

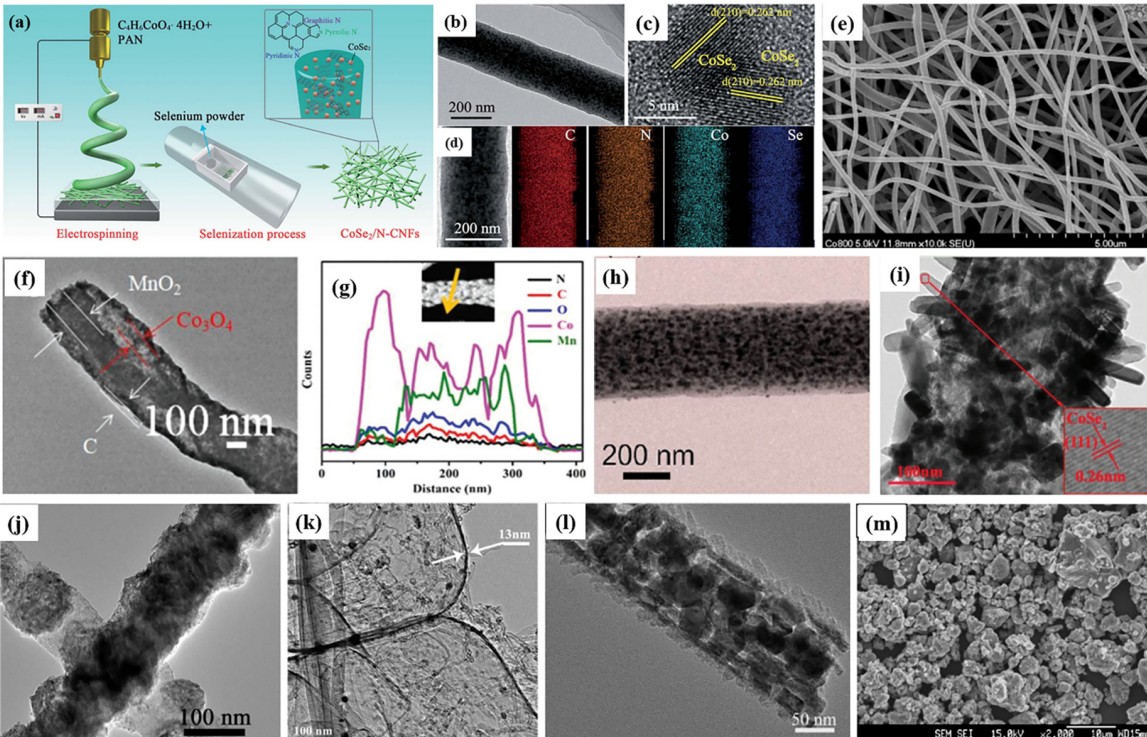

**Figure 8.** Preparation methods and morphology characterizations of cobalt-based composites. (**a**) Schematic illustration of the preparation of CoSe$_2$/N-carbon nanofibers (CNFs). (**b**) TEM image, (**c**) HRTEM image and (**d**) elemental mapping images of CoSe$_2$/N-CNFs [24]. (**e**) SEM image of Co-Sn/CNFs at 800 °C [90]. (**f**) TEM image and (**g**) HAADF-STEM-EDS line-scanning elemental mapping of Co$_3$O$_4$/MnO$_2$@C [91]. (**h**) TEM image of SnCo/PAN-CNFs [92]. (**i**) HRTEM image is taken from the selected zone [93]. (**j**) TEM image of Co/Co$_3$O$_4$-CNFs-700 [94]. (**k**) TEM image of the SnCo@CNT-3DC composite [44]. (**l**) SEM image of Co$_3$O$_4$ nanowire arrays [95]. (**m**) SEM image of the Co$_2$MnSi powder [96].

Chemical vapor deposition (CVD) processes have also been applied to synthesize cobalt-based electrode materials with multifarious structures [44,98–100]. He's group successfully prepared a novel structure via the CVD process [44]. The TEM image in Figure 8k clearly indicates a typical 0D-1D-3D hierarchical structure.

The magnetron sputtering method is also developed to prepare cobalt-based materials on substrates [19,101]. Chen et al. synthesized a Co$_3$O$_4$-C core-shell nanowire array via hydrothermal and magnetron sputtering. SEM image in Figure 8l shows that the Co$_3$O$_4$ nanowires were well-coated by carbon layers with a thickness of 18 nm.

Furthermore, Yoon's group reported an electric arc-melting method to produce a Co$_2$MnSi ingot [96]. Figure 8m indicates that the synthesized materials exhibited a certain degree of aggregation.

Various strategies for the preparation of cobalt-based electrode materials are summarized in Table 2. Obviously, hydrothermal/solvothermal, heat treatment, galvanic replacement and HEMM can be used to prepare different types of cobalt-based materials due to their economy and simplicity naturally. Other methods like electrospinning, electrodeposition and magnetron sputtering are more likely to synthesize cobalt-based materials with specific morphologies and structures, such as nanofibers and films. The diversity of the preparation process ensures the variety of synthesized cobalt-based electrode materials, thereby increasing the opportunities of cobalt-based materials for real batteries.

**Table 2.** Summary of the synthetic methods for cobalt-based electrode materials.

| Types of Materials | Synthetic Method | Co Source | Reference |
|---|---|---|---|
| $Co_3Sn_2@Co\text{-}NG$ | Hydrothermal (250 °C, 24 h) | $CoCl_2 \bullet 6H_2O$ | [53] |
| Special $CoSn_2/Sn$ alloy | Heat treatment (800 °C, Ar) | $Co_3O_4$ | [22] |
| Sn-Co-C ternary alloy | HEMM | Co powder | [34] |
| $Sn_{36}Co_{41}C_{23}$ alloy | Magnetron sputtering | Co powder | [19] |
| $Sn_{40}Co_{40}C_{20}$ | HEMM | Co powder | [45] |
| Sn-Co-C composite | HEMM | $Co(C_5H_7O_2)_3$ | [42] |
| Sn-Co alloy | Electrodepositing | $CoCl_2 \bullet 6H_2O$ | [43] |
| Sn-Co/CNTs | Galvanic replacement | $CoCl_2 \bullet 6H_2O$ | [102] |
| Sn-Co-C composite | HEMM | CoO | [83] |
| Nano-sized Co-Sn alloy | Hydrothermal (160 °C,48 h) | $CoCl_2 \bullet 6H_2O$ | [55] |
| $Sn_{30}Co_{30}C_{40}$ alloy | Ball-milling | Co powder | [81] |
| $CoSn_3$ | Hydrothermal (195 °C, 1.5 h) | $CoCl_2 \bullet 6H_2O$ | [56] |
| Sn-Co alloy | Electrodepositing | $CoCl_2 \bullet 6H_2O$ | [38] |
| $CoSn_2$ | Electrodepositing | Co electrode | [89] |
| Nanostructure Sn-Co-C composites | Galvanic replacement | $Co(CH_3COO)_2 \bullet 4H_2O$ | [103] |
| porous Sn-Co alloy | Electrodepositing | $CoCl_2 \bullet 6H_2O$ | [86] |
| $CoSn_3$-MWCNTs | Galvanic replacement | $CoCl_2 \bullet 6H_2O$ | [104] |
| $Sn_{54}Sb_{41}Co_5$ alloy | Electrodepositing | $CoCl_2 \bullet 6H_2O$ | [85] |
| $CoSn_5$ nanospheres | Galvanic replacement | $CoCl_2 \bullet 6H_2O$ | [105] |
| Grapheme wrapped-SnCo nanoparticles | Galvanic replacement | $CoCl_2 \bullet 6H_2O$ | [106] |
| $CoSn_2$ alloy | Electrodepositing | $CoCl_2 \bullet 6H_2O$ | [84] |
| Sn-Co-CNT@CNT | Galvanic replacement | $CoCl_2 \bullet 6H_2O$ | [67] |
| Co-Sn/C nanofiber | Electrodepositing | $C_4H_6CoO_4$ | [90] |
| SnCo/C nanofibers | Electrospinning | $C_4H_6CoO_4$ | [92] |
| Sn-Co@PMMA nanospheres | Galvanic replacement | $C_4H_6CoO_4$ | [40] |
| Amorphous $Co_3Sn_2$ | Hydrothermal (180 °C, 24 h) | $CoCl_2 \bullet 6H_2O$ | [107] |
| SnCo@CNT-3DC | CVD | $CoCl_2 \bullet 6H_2O$ | [44] |
| Sn-Co@C | Galvanic replacement | $Co(NO_3)_2 \bullet 6H_2O$ | [64] |
| carbon encapsulated Sn-Co alloy | Heat treatment (800 °C, Ar/H$_2$) | $Co_3O_4$ | [41] |
| $NiCo_2O_4$ powders | Heat treatment (320 °C, air) | $CoC_2O_4$ | [72] |
| $Co_2MnSi$ | Arc-melting | Co powder | [96] |
| Sb-Co-P | Electroplating | $CoCl_2 \bullet 6H_2O$ | [50] |
| CoSnC alloy | Galvanic replacement | $CoSO_4 \bullet 6H_2O$ | [108] |
| $Co_2SnO_4$ HC@rGO | Heat treatment (900 °C, Ar) | $CoCl_2 \bullet 6H_2O$ | [39] |
| $CoSnO_3$/GN/CNTs | Galvanic replacement | $CoCl_2 \bullet 6H_2O$ | [109] |
| $CoMn_2O_4$ | Hydrothermal (180 °C,10 h) | $Co(CH_3COO)_2 \bullet 4H_2O$ | [51] |
| $CoSnO_3$ | Galvanic replacement | $CoCl_2 \bullet 6H_2O$ | [52] |
| $Co_3O_4$ | Heat treatment (800 °C, Air) | $CoCO_3$ | [15] |

**Table 2.** *Cont.*

| Types of Materials | Synthetic Method | Co Source | Reference |
|---|---|---|---|
| $Co_3O_4$ nanotubes | Heat treatment (500 °C, Oxygen) | $Co_4(CO)_{12}$ | [73] |
| $Co_3O_4$ thin films | Electrodepositing | $Co(NO_3)_2$ | [88] |
| porous $Co_3O_4$ thin films | Electrodepositing | $Co(NO_3)_2$ | [87] |
| $Co_3O_4$ microspheres | Hydrothermal (200 °C,3 h) | $Co(NO_3)_2 \bullet 6H_2O$ | [21] |
| needlelike $Co_3O_4$ nanotubes | Galvanic replacement | $Co(NO_3)_2 \bullet 6H_2O$ | [65] |
| $Co_3O_4$ nanosheets | Hydrothermal (140 °C,20 h) | $CoCl_2 \bullet 6H_2O$ | [54] |
| agglomerated $Co_3O_4$ | Heat treatment (600 °C, Air) | $Co_3(NDC)_3(DMF)_4$ | [74] |
| macroporous $Co_3O_4$ platelets | Heat treatment (450 °C, Air) | $CoCl_2 \bullet 6H_2O$ | [75] |
| $Co_3O_4$/graphene | Galvanic replacement | $(C_2H_3O_2)_2Co \bullet 4H_2O$ | [110] |
| $Co_3O_4$/graphene | Galvanic replacement | $CoCl_2 \bullet 6H_2O$ | [111] |
| $Co_3O_4$ porous nanocages | Heat treatment (400 °C, Air) | $(C_2H_3O_2)_2Co \bullet 4H_2O$ | [76] |
| single-crystalline $Co_3O_4$ nanobelts | Hydrothermal (120 °C, 24 h) | $Co(NO_3)_2 \bullet 6H_2O$ | [57] |
| C@$Co_3O_4$ | Hydrothermal (180 °C, 8 h) | $Co(NO_3)_2 \bullet 6H_2O$ | [112] |
| $Co_3O_4$ nanorods/graphene nanosheets | Hydrothermal (120 °C, 12 h) | $CoSO_4 \bullet 7H_2O$ | [113] |
| mesoporous $Co_3O_4$ | Galvanic replacement | $Co(NO_3)_2 \bullet 6H_2O$ | [114] |
| $Co_3O_4$ nanocages | Heat treatment (550 °C, Air) | $(C_2H_3O_2)_2Co \bullet nH_2O$ | [77] |
| graphene/$Co_3O_4$ | Hydrothermal (80 °C, 4 h) | $CoCl_2 \bullet 6H_2O$ | [115] |
| CoO/graphene | Heat treatment (350 °C, Ar/$H_2$) | $Co(NO_3)_2 \bullet 6H_2O$ | [78] |
| $Co_3O_4$/graphene | Galvanic replacement | $(C_2H_3O_2)_2Co \bullet 4H_2O$ | [116] |
| multi-shelled $Co_3O_4$ hollow microspheres | Heat treatment (500 °C, Air) | $Co(Ac)_2 \bullet 4H_2O$ | [69] |
| shale-like $Co_3O_4$ | Heat treatment (400 °C, Air) | $(C_2H_3O_2)_2Co \bullet 4H_2O$ | [35] |
| hierarchical $Co_3O_4$/CNTs | Galvanic replacement | $Co(NO_3)_2 \bullet 6H_2O$ | [63] |
| bowl-like hollow $Co_3O_4$ microspheres | Heat treatment (400 °C, Air) | $(C_2H_3O_2)_2Co \bullet 4H_2O$ | [117] |
| mesoporous $Co_3O_4$ nanoflakes | Heat treatment (250 °C, Air) | $CoCO_3$ | [118] |
| hollow structured $Co_3O_4$ nanoparticles | Heat treatment (400 °C, Air) | $CoCl_2 \bullet 6H_2O$ | [119] |
| $Co_3O_4$/MCNTs | Heat treatment (500 °C, Air) | $CoCO_3$ | [120] |
| peapod-like $Co_3O_4$@carbon nanotube | Heat treatment (450 °C, Ar) | $Co(NO_3)_2 \bullet 6H_2O$ | [121] |
| $Co_3O_4$/CNT nanocomposites | Heat treatment (300 °C, Air) | $CoCl_2 \bullet 6H_2O$ | [122] |
| layer-by-layer $Co_3O_4$/graphene | Hydrothermal (170 °C, 15 h) | $(C_2H_3O_2)_2Co \bullet 4H_2O$ | [123] |
| $Co_3O_4$/CNTs nanotubes | Hydrothermal (120 °C, 2 h) | $(C_2H_3O_2)_2Co \bullet 4H_2O$ | [62] |
| Co/$Co_3O_4$ nanoparticles | Electrospinning | $CoCl_2 \bullet 6H_2O$ | [94] |
| $Co_3O_4$@NC | Heat treatment (550 °C, Ar) | ZIF-67 | [124] |
| carbon doped $Co_3O_4$ hollow nanofibers | Hydrothermal (180 °C, 12 h) | $Co(NO_3)_2 \bullet 6H_2O$ | [125] |
| Ni-doped Co/CoO/NC hybrid | Heat treatment (500 °C, Ar) | $Co(NO_3)_2 \bullet 6H_2O$ | [126] |

**Table 2.** *Cont.*

| Types of Materials | Synthetic Method | Co Source | Reference |
|---|---|---|---|
| hollow $Co_3O_4$/NGC | Heat treatment | $Co(NO_3)_2 \bullet 6H_2O$ | [127] |
| starfish-like $Co_3O_4$@nitrogen-doped carbon | Heat treatment | $Co(NO_3)_2 \bullet 6H_2O$ | [128] |
| yolk-shell $Co_3O_4$/C dodecahedrons | Heat treatment (350 °C, Air) | $Co(NO_3)_2 \bullet 6H_2O$ | [129] |
| hexagonal $Co_3O_4$ nanosheets | Hydrothermal (120 °C, 10 h) | $Co(NO_3)_2 \bullet 6H_2O$ | [130] |
| ultrathin mesoporous $Co_3O_4$ nanosheet | Heat treatment (450 °C, Air) | $Co(NO_3)_2 \bullet 6H_2O$ | [131] |
| flower-like $Co_3O_4$/C nanosheets | Heat treatment (500 °C, Air) | $Co(NO_3)_2 \bullet 6H_2O$ | [132] |
| $Co_3O_4$/GF | Heat treatment (300 °C, Air) | $Co(NO_3)_2 \bullet 6H_2O$ | [133] |
| ES-CN$Co_3O_4$ fibers | Heat treatment (800 °C, Ar/$H_2$) | $Co(NO_3)_2 \bullet 6H_2O$ | [11] |
| $Co_3O_4$/$MnO_2$@C | Electrospinning | $(C_2H_3O_2)_2Co \bullet 4H_2O$ | [91] |
| $CoP_3$ | Electrodeposition | CoO | [18] |
| $CoP_x$ | Ball milling | Co powder | [82] |
| $Co_2P$ | Electrodeposition | $CoCl_2 \bullet 6H_2O$ | [46] |
| $Co_xP$ | Heat treatment (320 °C, Ar) | $(C_2H_3O_2)_2Co \bullet 4H_2O$ | [79] |
| $Co_xP$ | HEMM | Co powder | [134] |
| CoP/RGO | Hydrothermal (180 °C, 16 h) | $CoCl_2 \bullet 6H_2O$ | [135] |
| CoP microflake | Heat treatment (350 °C, Ar) | $Co(NO_3)_2 \bullet 6H_2O$ | [136] |
| CoP nanorod | Hydrothermal (100 °C, 12 h) | $Co(NO_3)_2 \bullet 6H_2O$ | [61] |
| honeycomb-like CoP/$Co_2P$ | Heat treatment (600 °C, Ar) | $CoCl_2 \bullet 6H_2O$ | [137] |
| core-shell CoP/FeP porous microcubes | Heat treatment (300 °C, Ar) | $(C_2H_3O_2)_2Co \bullet 4H_2O$ | [70] |
| mesoporous CoP nanorods | Hydrothermal (120 °C, 6 h) | $Co(NO_3)_2 \bullet 6H_2O$ | [138] |
| $Co_2P$-Co/graphene | Galvanic replacement | $(C_2H_3O_2)_2Co \bullet 4H_2O$ | [139] |
| A-$Co_2P$/$C_xN_yB_z$-650 | Heat treatment (650 °C, Ar) | $Co(NO_3)_2 \bullet 6H_2O$ | [140] |
| CoP/CNS | Hydrothermal (120 °C, 6 h) | $Co(NO_3)_2 \bullet 6H_2O$ | [13] |
| CoP@NPPCS | Heat treatment (900 °C, Ar) | $(C_2H_3O_2)_2Co \bullet 4H_2O$ | [141] |
| CoP hollow nanorods/graphene | Hydrothermal (140 °C, 10 h) | $CoCl_2 \bullet 6H_2O$ | [25] |
| carbon-encapsulated CoP nanoparticles | Heat treatment (850 °C, Ar) | $Co(NO_3)_2 \bullet 6H_2O$ | [142] |
| $Co_2P$@N-C@rGO | Heat treatment (900 °C, Ar) | $Co(NO_3)_2 \bullet 6H_2O$ | [143] |
| $CoP_3$@Ppy microcubes | Coprecipitation& Parkerizing | $(C_2H_3O_2)_2Co \bullet 4H_2O$ | [144] |
| carbon coated $CoP_3$ | HEMM | Co powder | [80] |
| $Co_9S_8$@C nanoparticles | Hydrothermal & heat treatment | $Co(NO_3)_2 \bullet 6H_2O$ | [23] |
| 3D spongy $CoS_2$ nanoparticles/carbon | Freeze-dry &heat treatment &hydrothermal | $Co(NO_3)_2 \bullet 6H_2O$ | [12] |
| hollow $Co_9S_8$@C | Heat treatment & sulfuration | $(C_2H_3O_2)_2Co \bullet 4H_2O$ | [145] |
| CoS@S-doped OLC | Hydrothermal & heat treatment | $Co(NO_3)_2 \bullet 6H_2O$ | [146] |
| $SnS_2$@$CoS_2$-rGO | Hydrothermal (180 °C, 24 h) | $CoCl_2 \bullet 6H_2O$ | [147] |
| $CoS_2$@NCH | Solvothermal & heat treatment | $Co(NO_3)_2 \bullet 6H_2O$ | [148] |
| $CoS_2$/C micropolyhedron | Heat treatment (900 °C, $N_2$) | $Co(NO_3)_2 \bullet 6H_2O$ | [33] |

**Table 2.** *Cont.*

| Types of Materials | Synthetic Method | Co Source | Reference |
|---|---|---|---|
| CoS-24 | Solvothermal (180 °C 24 h) | $CoCl_2 \bullet 6H_2O$ | [149] |
| CNT@NC@CuCo$_2$S$_4$ | Solvothermal (200 °C 12 h) | $(C_2H_3O_2)_2Co \bullet 4H_2O$ | [59] |
| Co$_9$S$_8$-QDs@NC | Heat treatment (750 °C, Ar) | $CoCl_2 \bullet 6H_2O$ | [30] |
| TiO$_2$ nanobelts@Co$_9$S$_8$ | Hydrothermal & heat treatment | $(C_2H_3O_2)_2Co \bullet 4H_2O$ | [150] |
| 7-CoS/C | Heat treatment & sulfur | $Co(NO_3)_2 \bullet 6H_2O$ | [151] |
| MWCNTs/Co$_9$S$_8$ composites | Solvothermal & heat treatment | $Co(NO_3)_2 \bullet 6H_2O$ | [152] |
| Co$_9$S$_8$/Co | Ball-milling | Co powder | [153] |
| Co$_9$S$_8$@CNNs | Freeze-drying & heat treatment | $CoCl_2 \bullet 6H_2O$ | [154] |
| CoS$_2$/NCNTF | Heat treatment & sulfur | $Co(NO_3)_2 \bullet 6H_2O$ | [155] |
| Co$_9$S$_8$/N-C hollow nanospheres | Heat treatment & sulfur | $CoSO_4 \bullet 7H_2O$ | [28] |
| Co$_9$S$_8$/RGO | Hydrothermal (180 °C, 12 h) | $(C_2H_3O_2)_2Co \bullet 4H_2O$ | [97] |
| CoS$_2$/G composite | Hydrothermal (200 °C, 12 h) | $(C_2H_3O_2)_2Co \bullet 4H_2O$ | [156] |
| CoS$_2$@MCNF | Hydrothermal & heat treatment | $CoCl_2 \bullet 6H_2O$ | [157] |
| Ni$_3$S$_2$/Co$_9$S$_8$/N-doped carbon composite | Hydrothermal & heat treatment | $Co(NO_3)_2 \bullet 6H_2O$ | [158] |
| CoSe/Co@NC | Heat treatment (800 °C, Ar) | $Co(NO_3)_2 \bullet 6H_2O$ | [159] |
| Co$_{0.85}$Se NSs/G | Hydrothermal (180 °C, 16 h) | $(C_2H_3O_2)_2Co \bullet 4H_2O$ | [160] |
| Co-Zn-Se@C | Hydrothermal & heat treatment | $Co(NO_3)_2 \bullet 6H_2O$ | [161] |
| CoSe$_2$@NC-NR/CNT | Selenization | $Co(NO_3)_2 \bullet 6H_2O$ | [162] |
| cobblestone-like CoSe$_2$@C nanospheres | Heat treatment (500 °C, Ar/H$_2$) | $Co(NO_3)_2 \bullet 6H_2O$ | [32] |
| Cu-doped CoSe$_2$ microboxes | Hydrothermal (160 °C, 8 h) | $(C_2H_3O_2)_2Co \bullet 4H_2O$ | [163] |
| CoSe$_2$/N-CNFs | Electrospinning & heat treatment | $(C_2H_3O_2)_2Co \bullet 4H_2O$ | [24] |
| CoSe$_2$/C-ND@RGO | Heat treatment (600 °C, Air) | $Co(NO_3)_2 \bullet 6H_2O$ | [164] |
| CoSe@CSs | Hydrothermal & heat treatment | $Co(NO_3)_2 \bullet 6H_2O$ | [60] |
| CoSe quasi-microspheres | Hydrothermal (180 °C, 2 h) | $Co(NO_3)_2 \bullet 6H_2O$ | [165] |
| yolk-shell structured CoSe/C | Heat treatment (800 °C, Ar) | $Co(NO_3)_2 \bullet 6H_2O$ | [166] |
| Co$_9$Se$_8$/RGO hybrid nanosheet | Hydrothermal (180 °C, 12 h) | $(C_2H_3O_2)_2Co \bullet 4H_2O$ | [97] |
| urchin-like CoSe$_2$ nanorods | Electrospinning & heat treatment | $(C_2H_3O_2)_2Co \bullet 4H_2O$ | [93] |
| CoSe$_2$@C/CNTs | Heat treatment & selenization | $Co(NO_3)_2 \bullet 6H_2O$ | [48] |
| CoSe$_2$@N-PGC/CNTs | Heat treatment & selenization | $Co(NO_3)_2 \bullet 6H_2O$ | [167] |
| Co/(NiCo)Se$_2$ box in box structure | Selenization (270 °C 6 h) | $Co(NO_3)_2 \bullet 6H_2O$ | [71] |
| CoSe$_2$ powders | Hydrothermal (180 °C, 18 h) | $Co(NO_3)_2 \bullet 6H_2O$ | [168] |
| CoSe$_2$ nanoparticles | Hydrothermal (180 °C, 24 h) | $CoCl_2 \bullet 6H_2O$ | [169] |
| CoSe$_2$ microspheres | Heat treatment | $Co(NO_3)_2 \bullet 6H_2O$ | [170] |
| CoSe@PCP | Heat treatment & selenization | $Co(NO_3)_2 \bullet 6H_2O$ | [68] |

## 4. Application of Cobalt-Based Anode Materials in LIBs/SIBs

### 4.1. Cobalt-Based Alloys and Its Composites

As mentioned above, the electrochemical properties of cobalt-based alloys are mostly decided by metal components alloyed with cobalt, owing to the inactive nature with Li/Na of the cobalt ion.

As an inactive component of cobalt-based alloys, the presence of phase cobalt can alleviate the volume fluctuation as a protective matrix, thereby improving the electrochemical performances of cobalt-based alloys. Despite these features, the electrochemical properties of cobalt-based alloys are still disturbed by the pulverization and shedding of active materials during periodical cycles. Many cobalt-based alloys have been reported as anode materials for LIBs or SIBs with outstanding electrochemical properties owing to their unique structures and suitable compositions [40,55,67,81,83,92,104,105,171].

Shi et al. reported a novel structure with a Sn-Co alloy embedded in porous N-doped carbon microboxes that exhibited a superior electrochemical performance when used as anode materials for LIBs [64]. In Figure 9a, the Sn-Co@C-2 electrode exhibited the highest specific capacity at various current densities, from 0.1 to 2 A g$^{-1}$, indicating a superior rate performance. Furthermore, the Sn-Co@C-2 electrode delivered outstanding capacity retention when the current density went back to 0.1 A g$^{-1}$. Ex-situ XRD of the Sn-Co@C-2 electrode before and after cycles are shown in Figure 9b. CoSn2 disappeared after the first cycle, and the CoSn phase maintained well even after 100 cycles, suggesting a good reversibility of the Sn-Co@C-2 electrode.

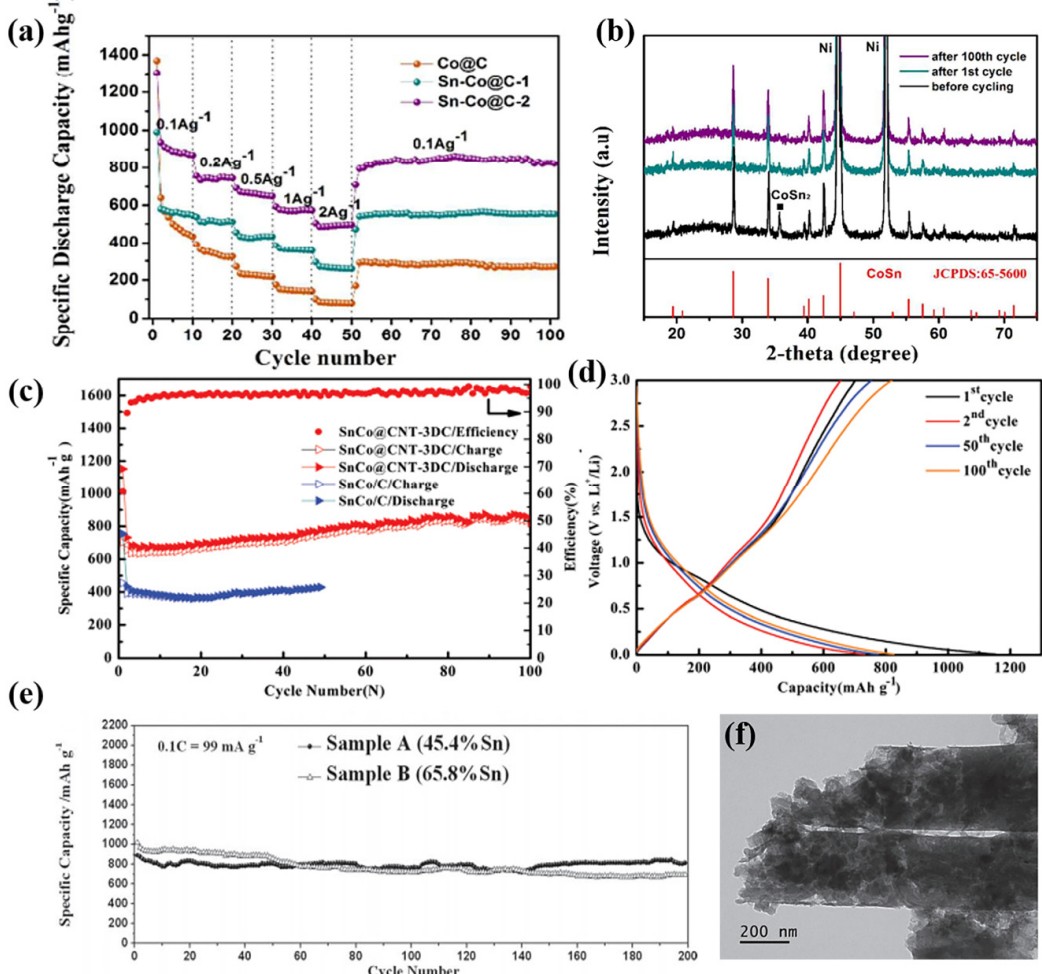

**Figure 9.** Electrochemical performances of cobalt-based anode materials. (**a**) Rate performances of Co@C, Sn-Co@C-1 and Sn-Co@C-2 as anode materials of LIBs. (**b**) XRD patterns of the Sn-Co@C-2 electrode before and after the charge-discharge cycles [64]. (**c**) Cycle performance and (**d**) charge/discharge profiles of the SnCo@CNT-3DC electrode at 50 mA g$^{-1}$ in the voltage range of 0.005–3 V vs. Li/Li$^+$ [44]. (**e**) Cycle performance of the Sn-Co-CNT@CNT nanostructure tested at 0.1 C (1C = 990 mA g$^{-1}$). (**f**) TEM image of the Sn-Co-CNT@CNT anode materials after 200 cycles at 1C in LIBs [67].

He's group reported a novel method for the one-step synthesis of hierarchical SnCo@CNT-3DC [44]. When used as anode materials for LIBs, this unique structure can provide numerous channels for electron transfer, shorten the diffusion pathway of Li$^+$ and ensure the permeation of electrolytes. As shown in Figure 9c, the SnCo@CNT-3DC electrode displayed a high capacity of 826 mA h g$^{-1}$ after 100 cycles at 0.1 A g$^{-1}$. Compared with the first discharge capacity, the capacity retention rate is 113%. During the cycles, there is a slight increase in capacity owing to the increasing electrochemical reaction of the SnCo nanoparticles caused by the activation of the electrode materials. As for the SnCo/C electrode, the relatively low capacity of 429.6 mA h g$^{-1}$ was retained after 50 cycles at 0.1 A g$^{-1}$. These results can be probably ascribed to the better transformation of lithium in the SnCo@CNT-3DC electrode. Figure 9d shows that the SnCo@CNT-3DC electrode delivered a high discharge capacity of 1154.4 mA h g$^{-1}$, with an initial coulombic efficiency (CE) of 61%. As the number of cycle increases, the capacity decreases slightly, suggesting good capacity retention.

Wang's group demonstrated an in-situ template technique for the preparation of the Sn-Co-CNT@CNT ternary tube-in-tube nanostructure [67]. When utilized as anode materials for LIBs, superior electrochemical properties were obtained owing to the confined volume change in the nanotube cavities and ensured a permanent electrical connectivity of the immobilized Sn-Co anodes. As shown in Figure 9e, the Sn-Co-CNT@CNT electrode with 45.4% Sn delivered a discharge capacity of 811 mA h g$^{-1}$ after 200 cycles, with 91.1% of the initial discharge capacity retained. The TEM image of the Sn-Co-CNT@CNT anode after 200 cycles demonstrates that the 1D nanotube-like structure was retained, indicating its structural stability during the cycles (Figure 9f).

## 4.2. Cobalt Oxides and Its Composites

Cobalt oxides have been identified as hopeful anode materials for LIBs and SIBs due to their high theoretical capacities, natural abundances, and low costs. For instance, Co$_3$O$_4$, as the most common cobalt oxide, has a theoretical capacity of 890 mA h g$^{-1}$ attributed to its eight-electron transfer reaction during cycling. However, the intrinsic defects of cobalt oxides, including low structural stability and poor electrical conductivity, lead to inferior electrochemical properties [57,63,77,78,111,113,116,118,120]. Yu's group reported hierarchically structured Co$_3$O$_4$@ carbon porous fibers to circumvent the volume fluctuation issue during lithiation/delithiation [11]. As shown in Figure 10a, the initial discharge and charge specific capacity of ES-CNCo$_3$O$_4$ was 1824 and 1003 mA h g$^{-1}$, with an initial columbic efficiency of 55%. After five cycles, the charge/discharge profiles were almost overlapped, indicating the high stability of the composite.

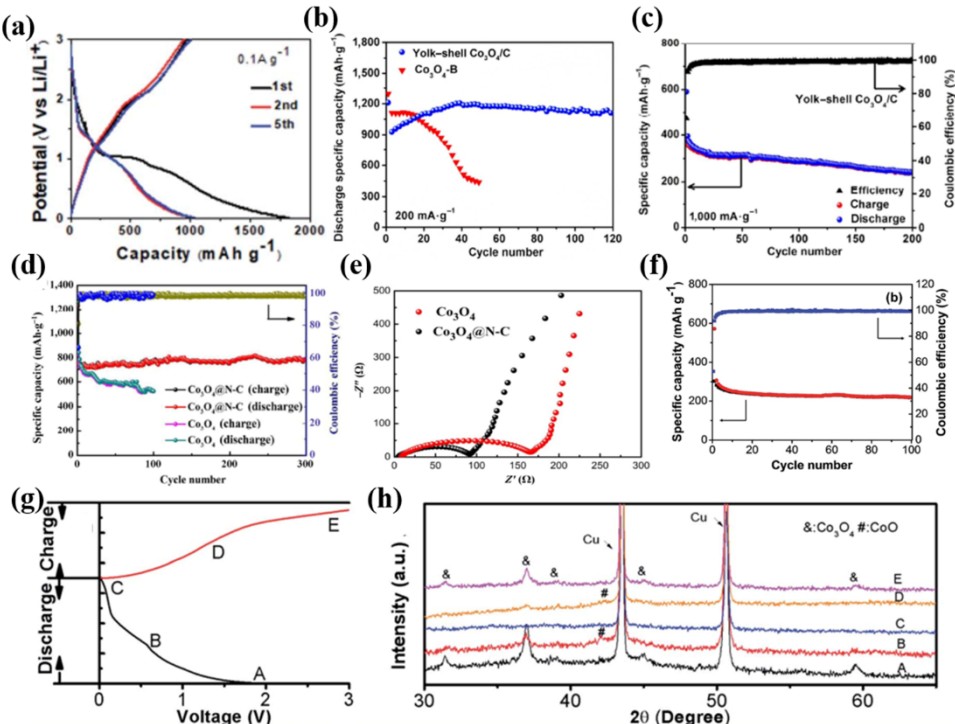

**Figure 10.** Electrochemical performances of cobalt-based anode materials. (**a**) Galvanostatic discharge-charge profiles of ES-CNCo$_3$O$_4$ at 0.1 A g$^{-1}$ [11]. (**b**) Cycling performance of the yolk-shell Co$_3$O$_4$ and Co$_3$O$_4$-B at 0.2 A g$^{-1}$ for LIBs. (**c**) Cycling properties of the yolk-shell Co$_3$O$_4$ at 1 A g$^{-1}$ for SIBs [129]. (**d**) Cycling performance of Co$_3$O$_4$ and Co$_3$O$_4$@N-C at 0.5 A g$^{-1}$. (**e**) Electrochemical impedance spectra for Co$_3$O$_4$ and Co$_3$O$_4$@N-C [128]. (**f**) Cycling performance of the Ni-doped Co/CoO/NC electrode and the corresponding columbic efficiencies at 0.5 A g$^{-1}$ [126]. (**g**) Galvanostatic charge-discharge profiles. Points A, B, C, D and E represent the pristine state, the discharge state at 0.6 V and 0.01 V and the charged state at 1.4 V and 3 V, respectively. (**h**) Ex-situ XRD patterns corresponding to these points [124].

Wu et al. synthesized yolk-shell Co$_3$O$_4$/C dodecahedrons to solve the problem of volume change during lithiation or sodiation [129]. When utilized as an anode for LIBs, the yolk-shell Co$_3$O$_4$/C electrode delivered a high discharge capacity of 1100 mA h g$^{-1}$ after 120 cycles. Regarding Co$_3$O$_4$-B, a lower specific capacity of 400 mA h g$^{-1}$ after 50 cycles was obtained, suggesting a better lithium storage performance of yolk-shell Co$_3$O$_4$/C. Figure 10c shows the sodium storage performance of yolk-shell Co$_3$O$_4$/C at a current density of 1 A g$^{-1}$. The second discharge capacity of the sample was 395 mA h g$^{-1}$, and the specific capacity of 240 mA h g$^{-1}$ was obtained after 200 cycles, suggesting the great electrochemical properties of yolk-shell Co$_3$O$_4$/C in SIBs.

Xie's group reported a porous starfish-like nanocomposite of N-doped carbon with Co$_3$O$_4$ coating [128]. The Co$_3$O$_4$@N-C electrode as a LIB anode delivered an initial specific capacity of 1235 mA h g$^{-1}$ and a capacity of 795 mA h g$^{-1}$ after 300 cycles, indicating a superior lithium storage performance to pure Co$_3$O$_4$ (Figure 10d). To better understand the gap between the electrochemical performances of the two samples, Nyquist plots of Co$_3$O$_4$@N-C and pure Co$_3$O$_4$ electrodes were fitted (Figure 10e). It is clear that the Co$_3$O$_4$@N-C composite exhibited smaller charge-transfer resistance, suggesting that the porous structure accelerates the diffusion of Li$^+$, and the presence of N-doped carbon improves the electrical conductivity of the sample, finally resulting in the superior electrochemical properties of Co$_3$O$_4$@N-C. Yang's group reported the preparation of a Ni-doped Co/CoO/N-doped carbon hybrid, with Ni-Co-ZIF as the precursor [126]. When utilized as an anode material for SIBs, the Ni-doped Co/CoO/NC electrode exhibited a high discharge capacity of 218.7 mA h g$^{-1}$ after 100 cycles, with 87.5% capacity retained (Figure 10f).

Wang and his collogues reported a nanocomposite of N-doped carbon with $Co_3O_4$ nanoparticles, exhibiting a superior sodium storage performance [124]. To further investigate the electrochemical mechanism of the electrode, five points that appeared at the first charge-discharge profile were tested by ex-situ XRD patterns (Figure 10g,h). Upon the discharge process, the phase of $Co_3O_4$ gradually disappeared along with the sodiation. When the cell discharged to 0.6 V (B), a new phase of CoO came out owing to the reduction of $Co_3O_4$. When fully discharged (C), only one phase of the Cu collector was retained, attributed to the poor crystallization of the $Na_2O$ phase. During the following charge process, reverse reactions occurred accompanied by the reappearance of CoO and $Co_3O_4$ at 1.4 V and only $Co_3O_4$ when fully charged (E). These results can be attributed to a reversible reaction of $Co_3O_4 + 8Na^+ + 8e^- \leftrightarrow 3Co + 4Na_2O$ during cycling.

### 4.3. Cobalt Sulfide and Its Composites

Cobalt sulfide has been widely studied as an anode material for LIBs/SIBs owing to its high theoretical capacity, satisfactory electrical conductivities and fair thermal stabilities. Similar to cobalt oxides, the main obstacle to the development of cobalt sulfide is the severe volume fluctuation during periodical cycling [14,16,17,26,36,47,49,66,172–186]. He and his colleagues synthesized self-assembled $CoS_2$ nanoparticles wrapped by $CoS_2$-quantum-dots-anchored graphene nanosheets (denoted as $CoS_2$ NP@G-$CoS_2$ QD) and tested as an anode material for LIBs [187]. As displayed in Figure 11a, the $CoS_2$ NP@G-$CoS_2$ QD delivered a high specific capacity of 831 mA h g$^{-1}$ at 1 A g$^{-1}$ after 300 cycles. Meanwhile, the nearly 100% coulombic efficiency over all 300 cycles also indicates the excellent cycle performance of this material. The enhanced cycle performance can be ascribed to the presence of few-layer graphene, which can not only improve the electrical conductivity of this composite but also alleviate the volume fluctuation and aggregation of $CoS_2$ (Figure 11b).

Sun's group synthesized a sponge-like composite assembled with $Co_9S_8$ quantum dots embedded in a carbon matrix, which was wrapped by rGO [172]. When utilized as an anode material for SIBs, the reaction mechanism was studied by an ex-situ XRD pattern in Figure 11c. Upon sodiation, the peaks of $Co_9S_8$ gradually disappeared until the discharge was completed. At the same time, the peaks of the $Na_2S$ and Co phases increasingly appeared owing to the reduction reaction from $Co_9S_8$ to Co and $Na_2S$. During the subsequent charge process, an inverse reaction, including the disappearance of the $Na_2S$ and Co phases and the reappearance of $Co_9S_8$, occurred. This situation indicates the high reversibility of $Co_9S_8$ quantum dots during the initial charge-discharge process. Peng's group built a CoS@rGO composite assembled by CoS nanoplates decorated on rGO and tested as an anode material for SIBs [173]. To investigate the industry application of this composite, a full cell was fabricated with $Na_3VPO_4$@C nanowires as the cathode (Figure 11d). As shown in Figure 11e, the reversible capacity of 290 mA h g$^{-1}$ was obtained after 100 cycles, suggesting the superior cycle performance of a CoS@rGO// $Na_3VPO_4$ (NVP)@C full battery. The alight red LED (inset of Figure 11e) fully demonstrates the practical application of a CoS@rGO//NVP@C full battery. This excellent electrochemical performance resulted from the novel structure and the presence of rGO.

Lou's group reported a $CoS_2$ nanobubble hollow prism and tested it as an anode material for LIBs [174]. This sample delivered remarkable specific capacities of 910, 778, 681 and 470 mA h g$^{-1}$ at the current densities of 200, 500, 1000 and 5000 mA g$^{-1}$, respectively (Figure 11f). It is worth mentioning that the capacity recovered to 864 mA h g$^{-1}$ when the current density was returned to 200 mA g$^{-1}$, indicating the outstanding robustness of the electrode. Qian's group synthesized hollow nanospheres assembled of mesoporous $Co_9S_8$ [175]. When utilized as an anode material of LIBs, an excellent specific capacity was maintained as 896 mA h g$^{-1}$ even after 800 cycles at 2 A g$^{-1}$ (Figure 11g). The superior cycle performance resulted from the remarkable charge-transfer kinetics. Furthermore, Zhou and coworkers prepared a one-dimensional multiwalled carbon nanotube@a-C@$Co_9S_8$ nanocomposites (MWCNT@a-C@$Co_9S_8$) as an anode material for advanced LIBs [5]. In Figure 11h, a high reversible capacity of 1065 mA h g$^{-1}$ was obtained after 700 cycles at a current density of 2 A g$^{-1}$. It is worth

mentioning that a slight fluctuation was observed during the cycling, ascribed to the enhanced surface lithium storage, the electrode reactivation and so on.

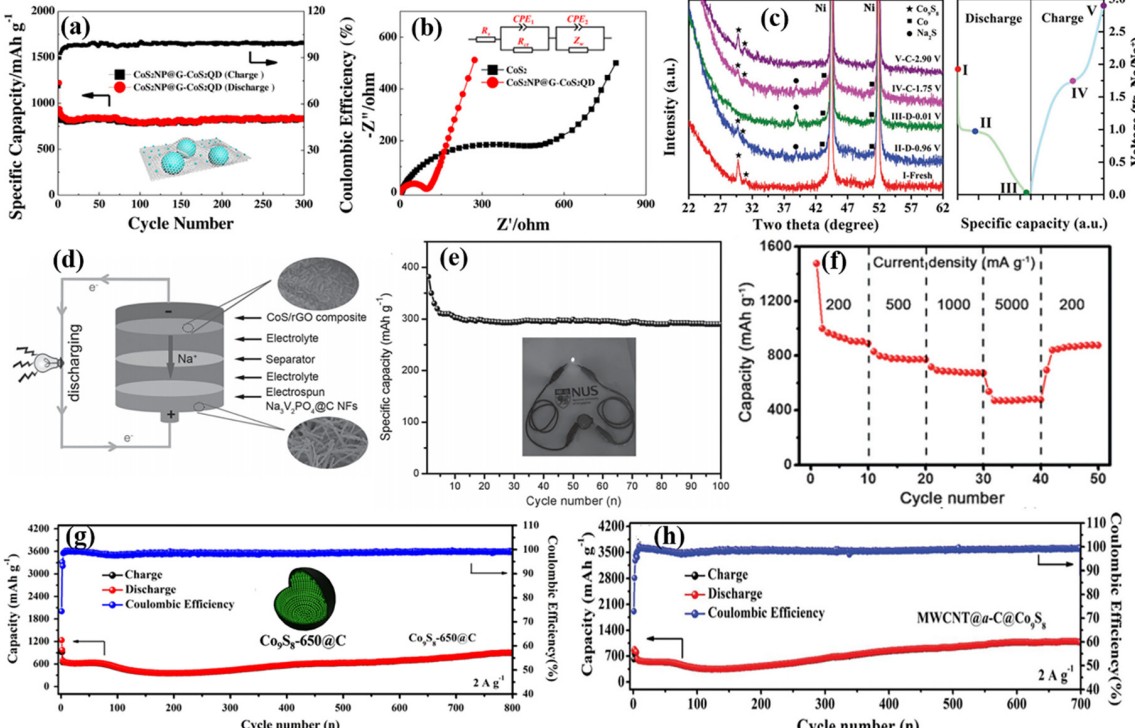

**Figure 11.** Electrochemical performances of cobalt-based anode materials. (**a**) Cycle performance of CoS₂-quantum-dots-anchored graphene nanosheets (CoS₂ NP@G-CoS₂ QD) at 1 A g⁻¹. (**b**) Nyquist plots of CoS₂ and CoS₂ NP@G-CoS₂ QD electrodes after 300 cycles [187]. (**c**) Ex-situ XRD of the cobalt sulfides quantum dots@mesoporous hollow carbon polyhedral@reduced graphene oxide (Co₉S₈ QD@HCP)@rGO electrode tested at different voltage states [172]. (**d**) Schematic illustration and (**e**) cycle performance of CoS@rGO//Na₃VPO₄ (NVP)@C sodium ion full cell [173]. (**f**) Rate performance of CoS₂ bubble-like hollow prisms [174]. (**g**) Cycle performance of Co₉S₈-650@C in LIBs [175]. (**h**) Cycle performance of Multiwalled carbon nanotube@a-C@Co₉S₈ nanocomposites (MWCNT@a-C@Co₉S₈) anode for LIBs [5].

## 4.4. Cobalt Phosphide and Its Composites

Cobalt phosphides, as promising electrode materials, have been studied as anode materials for LIBs/SIBs. Compared with cobalt oxides and sulfides, cobalt phosphides possess higher gravimetric/volumetric specific capacities. The main drawbacks that restrict their applications are the severe volume expansion during lithiation or sodiation and poor electrical conductivities. Many efforts have been made to circumvent these issues [18,31,46,82,134,136,137,188,189]. Sun's group hybridized the Co₂P nanoparticles with N-doped carbon matrices and tested the composite as an anode material for SIBs [143]. Except for the initial discharge profile, the profiles of the galvanostatic charge-discharge process overlapped well with each other, indicating a good reversibility of Co₂P@N-C@rGO (Figure 12a). Yan's group successfully fabricated CoₓP nanostructures with controlled phases, sizes and shapes. When applied as the anode material of LIBs, the hollow CoP particles delivered a capacity of 630 mA h g⁻¹ after 100 cycles, as shown in Figure 12b. With regard to solid CoP nanoparticles, a capacity of 480 mA h g⁻¹ was retained after 100 cycles. Better cycling performance of the hollow CoP electrode can be ascribed to the hollow structure, which ensures effective lithiation/delithiation and alleviates the volume change during periodical cycling.

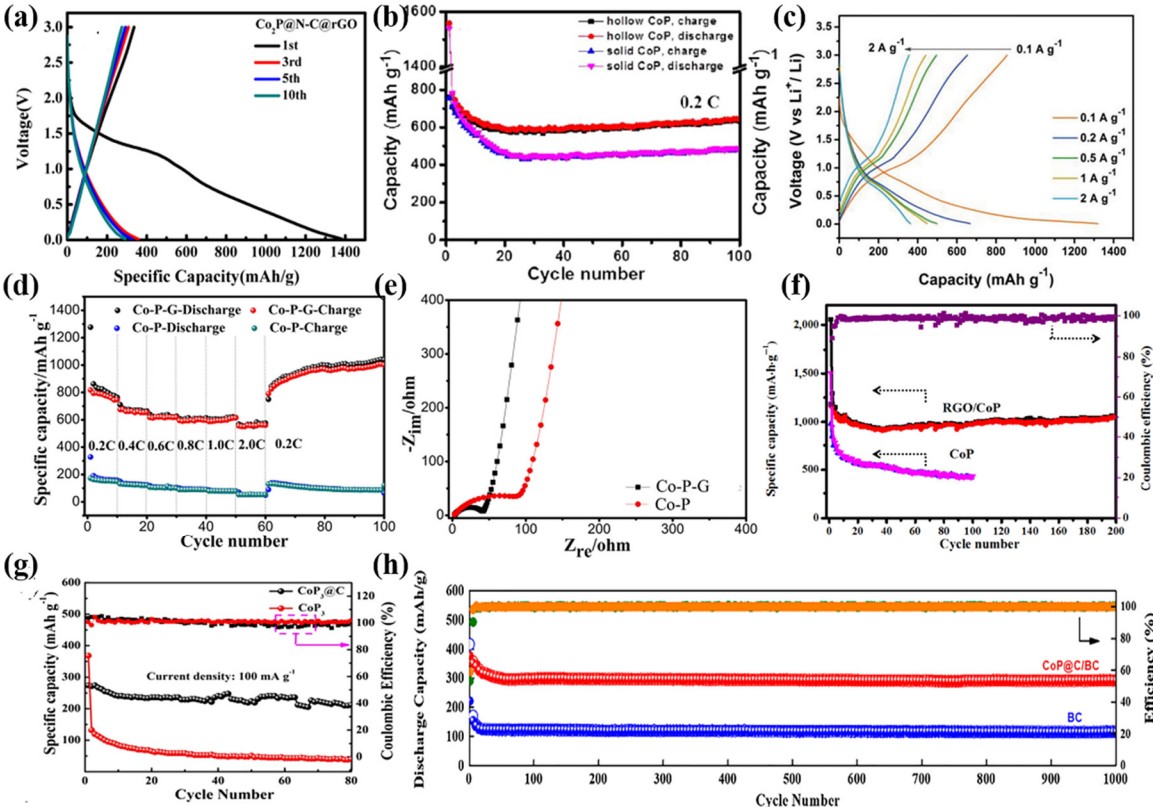

**Figure 12.** Electrochemical performances of cobalt-based anode materials. (**a**) Charge-discharge profiles of $Co_2P@$N-doped C (N-C)@reduced graphene oxide (rGO) in various cycles [143]. (**b**) Cycle performances of CoP hollow and solid nanoparticles [79]. (**c**) Charge-discharge profiles of N, P co-doped porous carbon sheet (CoP⊂NPPCS) hybrids at various current densities [141]. (**d**) Rate performances and (**e**) EIS spectra of $Co_2$P-Co/graphene and $Co_2$P-Co composites [139]. (**f**) Cycle performances of CoP/RGO and CoP [135]. (**g**) Cycle performances of $CoP_3@$C and $CoP_3$ [80]. (**h**) Cycling performance of CoP@C/biomass-derived carbon (BC) and pure BC [142].

Xiong's group firstly synthesized a novel composite with core-like CoP embedded in N, P co-doped porous carbon sheets (CoP⊂NPPCS) [141]. When utilized as an anode material of LIBs, the CoP⊂NPPCS electrode delivered reversible capacities of about 709, 500, 501, 410 and 357 mA h g$^{-1}$ at the current densities of 0.1, 0.2, 0.5 and 1.0 A g$^{-1}$, respectively. It is worth noting that the capacity retention was over 50% when the current density was expanded by 20 times, indicating the excellent rate capability. Peng's group prepared $Co_2$P-Co (Co-P) hollow nanospheres decorated with graphene sheets and tested them as anode materials for LIBs [139]. When the current densities increased from 0.2 C, 0.4 C, 0.6 C, 0.8 C and 1.0 C to 2.0 C (1C = 540 mA g$^{-1}$), the capacity of Co-P/graphene correspondingly decreased from 855, 674, 628, 608 and 610 to 567 mA h g$^{-1}$. When the current density returned to 0.2 C, a high capacity of 1039 mA h g$^{-1}$ was retained even after 100 cycles (Figure 12d). Clearly, the rate performance of Co-P/graphene was better than their $Co_2$P-Co counterparts, attributed to the presence of graphene, which improved the electrical conductivity of the electrode (Figure 12e).

Yang and co-workers synthesized a composite with cobalt phosphide nanowires and reduced the graphene oxide (CoP/RGO) [135]. As shown in Figure 12f, the cycle performances of CoP and CoP/RGO as anode materials of LIBs were displayed. The CoP/RGO electrode exhibited a higher capacity of 967 mA h g$^{-1}$ after 200 cycles. With regard to the CoP electrode, a capacity of only 429 mA h g$^{-1}$ was retained after 100 cycles. This situation can be ascribed to the framework of CoP/RGO, which possesses more free space in favor of the volume change during cycling. Zhang's group synthesized a carbon-coated $CoP_3$ nanocomposite and tested it as an anode material for SIBs [80]. The cycling performances of $CoP_3@$C and $CoP_3$ at 100 mA g$^{-1}$ are displayed in Figure 12g. The $CoP_3@$C

electrode exhibited a higher capacity than $CoP_3$ after 80 cycles, suggesting that the presence of the carbon layer contributed to the electrochemical performance of the electrode. Furthermore, Zhu's group successfully embedded ultrafine CoP nanoparticles into the carbon nanorod, which delivered a superior cycling performance when tested as an anode material of LIBs at 1 A $g^{-1}$ (Figure 12h) [142]. Well-utilized CoP was the main reason for this result.

### 4.5. Cobalt Selenide and Its Composites

Cobalt selenide, one of the transition metal chalcogenides, has attracted enormous attention in the field of anode materials for LIBs or SIBs owing to its good electrical conductivity and high theoretical capacity. Nevertheless, its poor structure stability during cycling is the main drawback of cobalt selenide [20,37,71,168–170,190–195]. As an anode material of LIBs, the $Co_{0.85}$Se NSs/G composite synthesized by Zhou's group delivered a high specific capacity of 730 mA h $g^{-1}$ after 300 cycles, as shown in Figure 13a [160]. With regard to $Co_{0.85}$Se NSs and $Co_{0.85}$Se NSs without a binder, the specific capabilities rapidly decreased to 50 mA h $g^{-1}$ after 50 cycles at 0.5 A $g^{-1}$. It should be noticed that the capacity of $Co_{0.85}$Se NSs/G has a slight increase during cycling, ascribed to the presence of graphene and the formation of a polymetric gel-like film. Moreover, the Na-ion storage cyclic performances of $Co_{0.85}$Se NSs/G, $Co_{0.85}$Se NSs and $Co_{0.85}$Se NSs without additives are displayed in Figure 13b [160]. Similar to lithium storage performances, the $Co_{0.85}$Se NSs/G electrode exhibited a higher reversible capacity of 193.8 mA h $g^{-1}$ after 100 cycles. The better lithium/sodium storage performance of the $Co_{0.85}$Se NSs/G composite can be attributed to its unique morphology and composition.

Pan's group successfully fabricated a composite with CoSe nanoparticles uniformly distributed in porous carbon polyhedral (CoSe@PCP) and tested as an anode material for LIBs and SIBs [68]. In Figure 13c, the CoSe@PCP electrode exhibited reversible capabilities of 701.2, 645.6, 590 and 457.5 mA h $g^{-1}$ when tested at 0.1, 0.5, 1 and 2 A $g^{-1}$, respectively. Moreover, the capacity could recover to 441.8, 452.7 and 524.1 mA h $g^{-1}$ when the current density was back to 1, 0.5 and 0.1 A $g^{-1}$, respectively. When utilized as an anode material for SIBs, the electrode delivered the capacities of 360.3, 315.6, 278.9, 247.1 and 207.7 mA h $g^{-1}$ tested at 0.05, 0.25, 1, 2 and 4 A $g^{-1}$, respectively. Furthermore, the capacities could recover to the original levels when the current densities returned to 0.5, 0.25 and 0.05 A $g^{-1}$ (Figure 13d). Clearly, the CoSe@PCP electrode shows excellent rate performances when applied as anode materials for LIBs and SIBs, attributed to the novel structure of CoSe@PCP.

Chen's group synthesized a novel urchin-like $CoSe_2$ and firstly used it as an anode material for SIBs [190]. As shown in Figure 13e, the $CoSe_2$ electrode delivered a high capacity of 0.410 A h $g^{-1}$ after 1800 cycles at 1 A $g^{-1}$, suggesting a superior cyclic performance. This result can be ascribed to the unique structure of the sample.

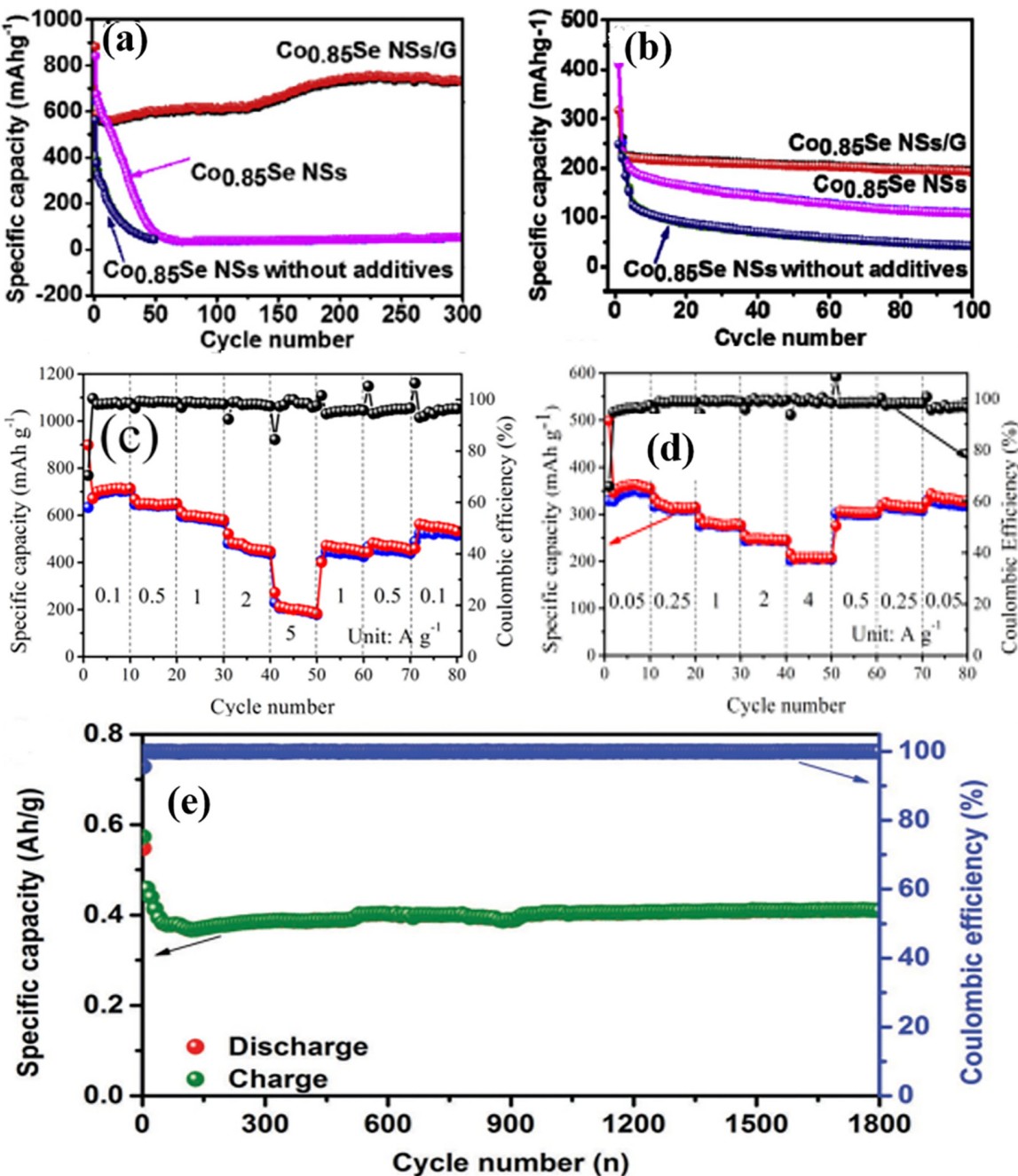

**Figure 13.** Electrochemical performances of cobalt-based anode materials. (**a**) Li-ion storage and (**b**) Na-ion storage cyclic performance of the $Co_{0.85}Se$ nanosheets/graphene (NSs/G) at 0.5 A $g^{-1}$, in comparison with $Co_{0.85}Se$ NSs and $Co_{0.85}Se$ NSs without additives [160]. Rate performance of CoSe@PCP for (**c**) LIBs and (**d**) SIBs [68]. (**e**) Cyclic properties of $CoSe_2$ at 1 A $g^{-1}$ [190].

## 4.6. Other Cobalt-Based Anode Materials

In addition to cobalt alloys, cobalt oxides, cobalt phosphide and cobalt chalcogenides, other cobalt-based anode materials like $CoMn_2O_4$ have also been utilized in LIBs and SIBs [39,50–52,72,96,108,109]. Zhang's group synthesized $CoMn_2O_4$ microspheres and evaluated them as anode materials for LIBs [51]. As displayed in Figure 14a, the $CoMn_2O_4$ electrode exhibited high initial charge and discharge capacities of 1860 and 2296 mA h $g^{-1}$ at 100 mA $g^{-1}$, respectively. During the first 200 cycles, the capacity gradually declined owing to a severe structure degradation caused by the conversion reaction.

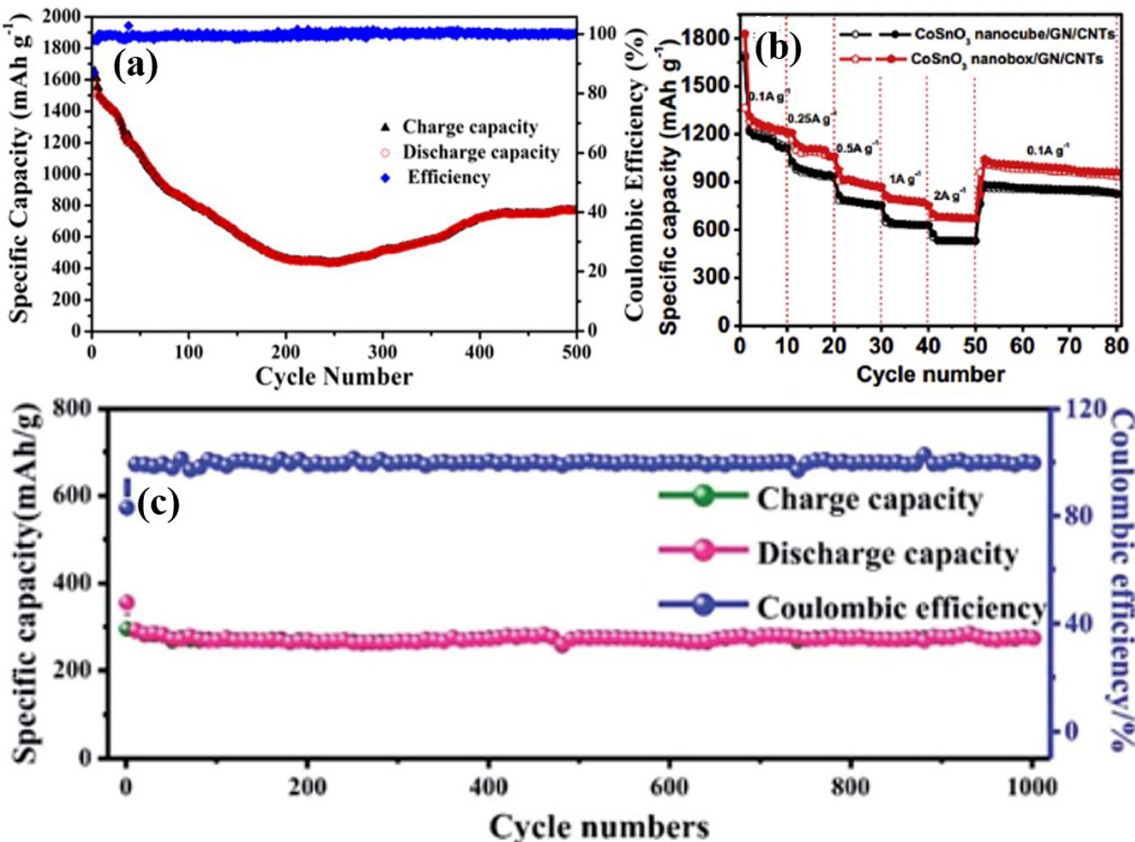

**Figure 14.** Electrochemical performances of cobalt-based anode materials. (**a**) Cyclic performance of $CoMn_2O_4$ microspheres at 100 mA g$^{-1}$ in LIBs [51]. (**b**) Rate performance of $CoSnO_3$ nanobox/graphene (GN)/carbon nanotubes (CNTs) and $CoSnO_3$ nanocube/GN/CNTs as anode materials of LIBs [109]. (**c**) Cyclic properties of $CoSnO_3$-N doped C (NC)-120 at 1 A g$^{-1}$ [52].

After 250 cycles, the reversible capacities increased, which can be ascribed to the gradual formation of the SEI and the degradation of the electrolyte. After 500 cycles, the $CoMn_2O_4$ electrode delivered a capacity of 722 mA h g$^{-1}$, indicating the enhanced performance of the $CoMn_2O_4$ electrode as an anode for LIBs. Wang et al. prepared $CoSnO_3$/GN/CNTs composite papers that delivered excellent electrochemical properties [109]. The rate performances of the $CoSnO_3$/GN/CNTs composite with different structures are displayed in Figure 14b. The $CoSnO_3$ nanobox/GN/CNTs composite delivered the capacities of 1233.8, 1106.4, 894.7, 780.3 and 676.7 mA h g$^{-1}$ at the current densities of 0.1, 0.25, 0.5, 1 and 2 A g$^{-1}$, respectively. Moreover, a high reversible capability of 992.8 mA h g$^{-1}$ could be retained when the current density returned to 0.1 A g$^{-1}$. The rate properties of $CoSnO_3$ nanocube/GN/CNTs were poorer than $CoSnO_3$ nanobox/GN/CNTs. Ji's group synthesized $CoSnO_3$-NCs nanoboxes and employed them as anode materials for SIBs [52]. The cyclic performance of this sample was tested at a current density of 1 A g$^{-1}$, as shown in Figure 14c. The $CoSnO_3$-NCs electrode exhibited a high capacity of 273.8 mA h g$^{-1}$ after 1000 cycles owing to the stable structure derived from the presence of the carbon layer.

The electrochemical properties of typical cobalt-based anode materials for LIBs and SIBs are summarized in Table 3. Clearly, this cobalt-based active material delivered different levels of rate performances and cyclic stabilities owing to the differences between morphology and composition. Hence, a rational structure is vitally important for the electrochemical properties of cobalt-based anode materials. Furthermore, the combination with carbonaceous materials is also an effective method to enhance the electrochemical properties of cobalt-based anode materials for LIBs and SIBs.

**Table 3.** Summary of the Li/Na-storage performances of typical cobalt-based anode materials. LIBs: lithium ion batteries and SIBs: sodium ion batteries.

| Types of Materials | Current Density (A g$^{-1}$) | Cut-off Voltage (V) | Cycle Number | Specific Capacity (mA h g$^{-1}$) | Reference |
|---|---|---|---|---|---|
| **For LIBs** | | | | | |
| Sn-Co@C-2 | 0.1 | 0.01–3 | 100 | 818 | [64] |
| SnCo@CNT-3DC | 0.1 | 0.005–3 | 100 | 826 | [44] |
| Sn-Co@PMMA | 0.1 | 0.001–2 | 100 | 594 | [40] |
| Co$_3$Sn$_2$@Co-NG | 0.25 | 0.005–3 | 100 | 1615 | [53] |
| SnCo/PAN-CNFs | 0.267 | 0.005–2.5 | 100 | 548 | [92] |
| Co-Sn/CNF-800 | 0.161 | 0.02–2.8 | 80 | 560 | [90] |
| Sn-Co-CNT@CNT | 0.099 | 0.005–3 | 200 | 811 | [67] |
| GNS-SnCo | 0.072 | 0.005–3 | 60 | 571 | [106] |
| Sb-Co-P | 0.1 | 0.02–1.5 | 50 | 539 | [50] |
| CoSnC | 0.1 | 0–2 | 50 | 450 | [108] |
| Co$_2$SnO$_4$ | 0.1 | 0.01–2.5 | 100 | 1000 | [39] |
| CoSnO$_3$/GN/CNTs | 0.1 | 0–3 | 150 | 1098.7 | [109] |
| CoMn$_2$O$_4$ | 0.1 | 0.001–3 | 500 | 772 | [51] |
| CoO$_x$/MCS | 0.07 | 0.01–3 | 30 | 703 | [100] |
| Co$_3$O$_4$/Graphene | 0.2 | 0.001–3 | 42 | 800 | [110] |
| Co$_3$O$_4$ nanobelts | 1 | 0–3 | 60 | 614 | [57] |
| Co$_3$O$_4$ nanorods/GNS | 1 | 0.01–3 | 40 | 1090 | [113] |
| Shale-like Co$_3$O$_4$ | 0.2 | 0.005–2.9 | 100 | 1045.3 | [35] |
| Co$_3$O$_4$ nanoflakes | 0.089 | 0.01–3 | 300 | 806 | [118] |
| Peapod-like Co$_3$O$_4$@CNT | 0.1 | 0–3 | 60 | 318 | [121] |
| CoP/C | 0.1 | 0–2 | 200 | 407 | [134] |
| CoP microflake | 1 | 0.01–3 | 800 | 619.2 | [136] |
| CoP/Co$_2$P | 0.2 | 0.01–3 | 450 | 851.2 | [137] |
| CoP nanorods | 0.5 | 0.01–3 | 300 | 894 | [138] |
| Co$_2$P-Co | 0.1 | 0.01–3 | 200 | 929 | [139] |
| CoP nanorod arrays | 0.4 | 0.01–3 | 900 | 390 | [61] |
| CoP HR@rGO | 0.1 | 0.005–3 | 100 | 714.7 | [25] |
| CoP@C/BC | 1 | 0.01–3 | 1000 | 351 | [142] |
| CoP$_3$@PPy | 0.5 | 0–2 | 220 | 650 | [144] |
| CoS$_2$NP@G-CoS$_2$QD | 1 | 0.05–3 | 300 | 831 | [187] |
| Worm-like CoS$_2$ | 0.1 | 0.01–3 | 100 | 883 | [180] |
| CoS$_2$@NG | 0.1 | 0.01–3 | 150 | 882 | [26] |
| NC/CoS$_2$-650 | 0.1 | 0.1–3 | 50 | 560 | [66] |
| Mesoporous Co$_9$S$_8$ | 2 | 0.01–3 | 800 | 896 | [175] |
| MWCNT@a-C@Co$_9$S$_8$ | 2 | 0.01–3 | 700 | 1065 | [5] |
| CoS$_2$ nanobubble | 1 | 0.05–3 | 200 | 737 | [174] |
| rGO/CoSe$_2$ | 0.2 | 0.01–3 | 200 | 1577 | [20] |
| Co$_{0.85}$Se nanosheets | 0.2 | 0.01–3 | 50 | 516 | [194] |
| Cu-doped CoSe$_2$ | 1 | 0.01–3 | 200 | 807 | [195] |
| CoSe@PCP | 1 | 0.005–3 | 500 | 708.2 | [68] |
| CoSe$_2$@C/CNTs | 1 | 0.5–2.9 | 1000 | 390 | [48] |
| CoSe$_2$@CNFs | 0.2 | 0.01–3 | 300 | 1405 | [93] |
| **For SIBs** | | | | | |
| SnCo@C | 0.1 | 0.1–2 | 120 | 276.2 | [41] |
| CoSnO3-NCs | 1 | 0.01–3 | 1000 | 273.8 | [52] |
| Shale-like Co$_3$O$_4$ | 0.05 | 0.005–2.9 | 50 | 380 | [35] |
| Co$_3$O$_4$@CNT | 0.05 | 0.01–3 | 100 | 403 | [122] |
| Co$_3$O$_4$@NC | 1 | 0.01–3 | 1100 | 175 | [124] |
| Ni-doped Co/CoO/NC | 0.5 | 0.01–3 | 100 | 218.7 | [126] |
| Yolk-shell Co$_3$O$_4$/C | 1 | 0.01–3 | 200 | 240 | [129] |
| Co$_3$O$_4$/MnO$_2$@C | 0.8 | 0.01–3 | 1000 | 126 | [91] |
| CoP@C-RGO-NF | 0.05 | 0.01–3 | 100 | 473.1 | [27] |
| RGO@CoP@C-FeP | 0.1 | 0.01–3 | 200 | 456.2 | [70] |
| A-Co$_2$P/C$_x$N$_y$B$_2$-650 | 0.2 | 0.005–2.5 | 100 | 251.2 | [140] |
| CoP nanorod arrays | 0.2 | 0.01–3 | 550 | 297 | [61] |
| CoP-O | 1 | 0.01–2.5 | 900 | 386 | [13] |
| Co$_2$P@N-C@rGO | 0.05 | 0.01–3 | 100 | 225 | [143] |
| CoP$_3$@C | 0.1 | 0–2.5 | 80 | 212 | [80] |

**Table 3.** *Cont.*

| Types of Materials | Current Density (A g$^{-1}$) | Cut-off Voltage (V) | Cycle Number | Specific Capacity (mA h g$^{-1}$) | Reference |
|---|---|---|---|---|---|
| CoS$_2$-MWCNT | 0.1 | 1–2.9 | 100 | 568 | [47] |
| cs-Co$_x$S$_y$/DPC | 0.5 | 0.01–3 | 50 | 300 | [181] |
| CoS$_2$/rGO | 1 | 0.01–3 | 1000 | 192 | [49] |
| CoS@rGO | 1 | 0.1–2.9 | 1000 | 420 | [173] |
| (Co$_9$S$_8$ QD@HCP)@rGO | 0.3 | 0.01–2.9 | 500 | 628 | [172] |
| CoS@rGO | 1 | 0.01–2.9 | 1000 | 420 | [173] |
| CoS$_x$@NSC | 1 | 0.01–3 | 200 | 606 | [186] |
| Co$_9$S$_8$@C | 5 | 0.01–3 | 1000 | 305 | [23] |
| Urchin-like CoSe$_2$ | 1 | 0.5–3 | 1800 | 410 | [190] |
| CoSe@PCP | 0.1 | 0.005–3 | 100 | 341 | [68] |
| CoSe$_2$ nanorods | 5 | 0.4–3 | 2000 | 386 | [168] |
| CoSe$_2$@N-PGC/CNTs | 0.2 | 0.001–3 | 100 | 424 | [48] |
| Co$_9$Se$_8$/rGO | 0.05 | 0.01–3 | 100 | 406 | [97] |
| CoSe/C | 0.5 | 0.01–3 | 50 | 552.5 | [166] |
| CoSe@100CSs | 4 | 0.5–2.8 | 10,000 | 260 | [60] |
| CoSe$_2$/N-CNF | 2 | 0.5–3 | 1000 | 308.4 | [24] |

## 5. Conclusions and Perspectives

The increasing consumption of fossil energy makes the development of clean energy necessary. Due to time and geographical constraints, high-performance RBs are urgent for the development of clean energy. Designing and synthesizing suitable electrode materials is critical for the development of RBs. This review systematically demonstrates that cobalt-based electrode materials are expected to be a new generation of anode materials for LIBs/SIBs.

In this review, we sequentially summarized the Li$^+$/Na$^+$ storage mechanism, typical preparation methods and applications of various cobalt-based anode materials, including cobalt-based alloy, cobalt oxide, cobalt sulfides and cobalt phosphides. For cobalt-based anode materials, their large volume changes and poor conductivity during charges and discharges are the main obstacles to the further development in LIBs/SIBs. In this regard, future research can focus on the following aspects:

(1) Developing a simple and feasible synthetic process for the preparation of cobalt-based anode materials with specific morphology and sizes, which can ensure satisfactory contact between the electrolyte and active material, and cycle performances of cobalt-based anode materials. In addition, the combination with the conductive material can improve the conductivity of the electrode material and improve the rate performances of the cobalt-based anode materials.

(2) More advanced characterization and calculation methods should be used to further study the Li$^+$/Na$^+$ storage mechanisms of cobalt-based anode materials, which will be meaningful for designing a suitable morphology.

(3) Considering commercial applications, the whole battery system, including cathode materials, binders, conductive agents, electrolytes and additives, should also be optimized. Appropriate matching materials can fully exploit the advantages of the high storage capacities of cobalt-based anode materials in LIBs/SIBs.

**Author Contributions:** Y.Z. and N.W. wrote the manuscript. Z.B. revised the manuscript. All authors commented on the manuscript and have contributed equally to this work. All authors have read and agreed to the published version of the manuscript.

**Funding:** This work was supported by the Australian Research Council (ARC; DE200101384) for the support provided through a Discovery Project (DP160102627) and a Linkage Project (LP160100273) and the Shanxi Province Science Foundation for Youths (No. 201901D211257).

**Acknowledgments:** We acknowledge the support from Australian Research Council for the support provided through a Discovery Project and a Linkage Project and the Shanxi Province Science Foundation for Youths.

**Conflicts of Interest:** The authors declare no conflict of interest.

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
