# Peer review of "The Progress of Cobalt-Based Anode Materials for Lithium Ion Batteries and Sodium Ion Batteries"

_applsci, doi:10.3390/app10093098_

Round 1
Reviewer 1 Report
Cobalt alloys and composites are attractive and promising anode materials for lithium ion batteries and sodium ion batteries. This manuscript reviews the reaction mechanisms, various synthetic methods, and the applications when cobalt-based materials are used for lithium ion batteries and sodium ion batteries. Overall, this manuscript is well structured and written, and is pretty informative. It could be interesting to the battery community. The following are some suggestions about revising the manuscript before it is considered for publication.
- The title uses the term “lithium/sodium ion batteries”. This is kind of confusing with the batteries that use lithium ions and sodium ions in a combination way. It is better to use “… anode materials for lithium ion batteries and sodium ion batteries”, which accurately reflects what the manuscript talks about, though it makes the title a little redundant.
- Except Figures 1 and 3, the rest figures have no a general caption, making the reader hard to catch the themes of the figures. For example, Figure 2 may use a general caption like “Li+ and Na+ storage in Sn-Co alloy anode.”, followed by “(a) …, (b) …, ”.
- Line 69, should be “graphene”, instead of “grapheme”?
- Line 82, “Moreover, we also conclude …” could be “Moreover, we also remarked …”.
- Line 93, use “during battery cycling”, instead of “during periodic cycles”.
- Line 111, in Eq. (2), “Sn +y Co” means two elements Sn and Co. However, as a rechargeable battery, the product after the charge process should be a “CoySn” alloy, in order to make the next round of discharge process take place.
- (4) has the same problem.
- While the first reaction equations were numbered, the rest ones should also be numbered sequentially to make the manuscript in a better shape.
- Lines 156 and 157, one more equation should be added to show how Na3P is converted to P prior to the next reaction happens.
- Lines 172 and 176, should be “discharge or charge process”, instead of “… rate”.
- Line 179, it is suggested to delete the sentence “we found that Co component has no electrochemical reactions with …”. It is meaningless because, according to the context, it is clear that the reactions occur between Co-based alloys or composites and lithium ions or sodium ions, instead of any reactions between Co and lithium ions or sodium ions. It doesn’t make sense to say a finding like this.
- Line 226, Figures 5e-g reveal …, instead of “Figure 5e-f reveal …”.
- Line 228, In Figures 5h-l, …
- Line 369, “Obviously, hydrothermal … are more common for …”. Please revise. This is illogical. Table 2 doesn’t provide any information that the methods are more common.
- Table 2 is arranged very untidily. It should be grouped according to the synthetic methods, instead of simply putting everything together.
- Table 3 should also be grouped according to the types of materials as described in the main text.
- Line 480, “… its composites”, instead of “its composite”.
Reviewer 2 Report
This paper summarises the work that has been performed on Cobalt based anode materials for both Li/Na systems.
The work is extensive and the references appropriate for this type of review. English level is also satisfactory.
Authors should also improve the print quality of the Figures should the paper be considered for publication.
